# Jacobian-Based Interpretation of Nonlinear Neural Encoding Model

**Xiaohui Gao**[1,*]**, Haoran Yang**[1,*]**, Yue Cheng**[2,*]**, Mengfei Zuo**[1]**, Yiheng Liu**[1]**, Peiyang Li**[3]**,**
**Xintao Hu**[1,†]

[1]Northwestern Polytechnical University    [2]Beijing Jiaotong University
[3]Chongqing University of Posts and Telecommunications
`gaitxh@foxmail.com, yhr0602@mail.nwpu.edu.cn, yuecheng@bjtu.edu.cn,`
`zuomengfei@mail.nwpu.edu.cn, liuyiheng123@mail.nwpu.edu.cn,`
`lipeiyang@cqupt.edu.cn, xhu@nwpu.edu.cn`
[*]Equal contribution    [†]Corresponding author

## Abstract

In recent years, the alignment between artificial neural network (ANN) embeddings and blood oxygenation level dependent (BOLD) responses in functional magnetic resonance imaging (fMRI) via neural encoding models has significantly advanced research on neural representation mechanisms and interpretability in the brain. However, these approaches remain limited in characterizing the brain's inherently nonlinear response properties. To address this, we propose the **J**acobian-based **N**onlinearity **E**valuation (JNE), an interpretability metric for nonlinear neural encoding models. JNE quantifies nonlinearity by statistically measuring the dispersion of local linear mappings (Jacobians) from model representations to predicted BOLD responses, thereby approximating the nonlinearity of BOLD signals. Centered on proposing JNE as a novel interpretability metric, we validated its effectiveness through controlled simulation experiments on various activation functions and network architectures, and further verified it on real fMRI data, demonstrating a hierarchical progression of nonlinear characteristics from primary to higher-order visual cortices, consistent with established cortical organization. We further extended **JNE** with **S**ample-**S**pecificity (JNE-SS), revealing stimulus-selective nonlinear response patterns in functionally specialized brain regions. As the first interpretability metric for quantifying nonlinear responses, JNE provides new insights into brain information processing. Code available at
https://github.com/Gaitxh/JNE.

## 1   Introduction

In recent years, researchers have made significant progress in constructing interpretable computational models of the brain by aligning artificial neural network (ANN) representations with functional magnetic resonance imaging (fMRI) signals [1, 2, 3, 4, 5, 6]. These studies have typically been based on the neural encoding model framework, wherein hierarchical embeddings from ANNs are used as input features to predict voxel-level neural responses (i.e., blood oxygenation level dependent responses in this study) using either linear models (e.g., ridge regression)[2, 7, 8, 9] or nonlinear regression models (e.g., deep neural networks)[10, 11, 5, 12], and the prediction performance is evaluated by metrics such as the coefficient of determination ($R^2$) or Pearson correlation coefficient ($r$)[11, 13, 14] (see Fig.1a). Based on this framework, a substantial body of work has shown that ANN-derived embeddings correspond well with the spatial distribution of functional brain regions in language [3, 4, 15, 16], vision [2, 17], and multimodal tasks [8, 18, 19]. With the advancement of neuroscience research, it has been observed that the brain's responses to external stimuli exhibit

39th Conference on Neural Information Processing Systems (NeurIPS 2025).

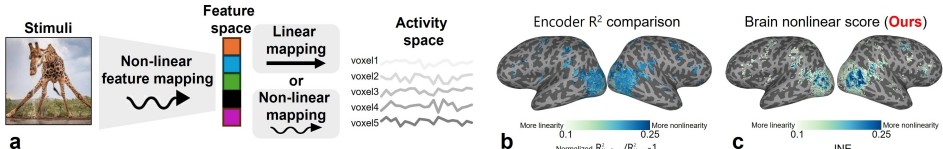

Figure 1: **Comparison of brain encoding frameworks and nonlinearity analysis approaches.**
**a**. Neural encoding aligns ANN representations with fMRI BOLD responses via linear/nonlinear models to predict voxel responses. **b**. Nonlinearity estimation via $R^2$ ratios between linear/nonlinear models fails when deep features embed nonlinearities, yielding comparable $R^2$ and poor regional discriminability. **c**. JNE characterizes BOLD nonlinearity by statistically measuring Jacobian dispersion across inputs, showing higher discriminability across regions.

marked nonlinear characteristics. Numerous studies have demonstrated the nonlinear nature of neural responses [10, 11, 20, 21, 22, 23, 24, 25, 26]. However, key questions such as how to quantitatively characterize the degree of blood oxygenation level dependent (BOLD) response nonlinearity and how it is distributed across the cortex remain insufficiently addressed by systematic studies and theoretical frameworks.

Current neural encoding models face fundamental limitations in terms of nonlinear interpretability: linear neural encoding models can only establish simple mappings and fail to capture higher-order nonlinear features, while nonlinear neural encoding models are difficult to interpret due to their inherent "black-box" nature [11]. Therefore, quantitatively interpreting the nonlinear characteristics of neural encoding models has become a critical challenge. A common strategy is to infer the degree of nonlinearity in a given brain region by comparing the prediction performance of linear and nonlinear neural encoding models in that region [27, 28, 29]. However, this strategy essentially reflects only performance differences between models and fails to accurately reveal the intrinsic nature of BOLD response nonlinearity (see Fig.1b and Section 3.2).

To address the above issues, we propose a novel interpretability metric for nonlinearity—**J**acobian-based **N**onlinearity **E**valuation (JNE)—designed to quantify the nonlinear characteristics of neural encoding models, thereby approximating voxel-level BOLD nonlinearity. The core idea is to calculate the local linear mappings (i.e., local derivatives, also known as Jacobian matrices) of the neural encoding model under different input stimuli, and to evaluate the level of nonlinearity by statistically measuring their variability across inputs (theoretical description in Section A.1). Intuitively, an ideal linear neural encoding model should maintain a constant mapping across different inputs, resulting in a JNE value of 0; in contrast, if the model exhibits significant nonlinearity, the Jacobian matrices will vary across inputs, yielding a higher JNE value. Centering on JNE, we validate via simulations for nonlinearity quantification, then apply to fMRI for approximating visual cortex BOLD nonlinearity. JNE serves as a conditional interpretability metric for approximating voxel-level BOLD nonlinearity, rather than a direct neural measure, due to BOLD confounds. In summary, the main contributions of this study include:

- We propose JNE as a novel interpretability metric for nonlinear neural encoding models, which quantifies nonlinearity by statistically measuring the dispersion of input-output Jacobian matrices (Section 2.4), and theoretical derivation indicates that JNE can serve as an approximate measure of BOLD response nonlinearity (Section A.8). Controlled simulation experiments validate the effectiveness of JNE in quantifying the nonlinearity of output responses (Sections 3.1 and A.7);

- Our results demonstrate that inferring nonlinear properties by comparing $R^2$ values between linear and nonlinear neural encoding models is insufficient to reveal the brain's nonlinear responses, especially when deep visual representations already embed nonlinear transformations (Section 3.2);

- We find that the primary visual cortex tends to exhibit more linear responses, while higher-order visual cortices show stronger nonlinear characteristics under natural visual stimulation (Section 3.3). Overall, a hierarchical structure emerges with increasing nonlinearity from the primary to intermediate to higher-order visual cortex (Section 3.4);

- We further define JNE-SS (Section A.10), which characterizes the nonlinear properties of individual samples at specific voxels. Our results indicate that BOLD response nonlinearity exhibits sample-selective preferences (Section 3.5).

## 2 Methods

### 2.1 fMRI Dataset Description and Feature Extraction

In this study, we used the Natural Scenes Dataset (NSD) [30], which provides whole-brain 7T fMRI responses from multiple subjects while viewing natural scene images. We focused on four subjects (S1, S2, S5, S7) who completed the full experimental protocol as the primary participants in our analysis. Each image and its corresponding averaged beta map were treated as a single sample. The dataset was split into training, validation, and testing sets with a ratio of 8:1:1 (8000:1000:1000 samples), ensuring no subject overlap between the training and validation sets, while the test set included repeated samples across subjects. We employed the pre-trained CLIP-ViT [31][1] image encoder to obtain computational representations of the visual stimuli (Fig. 2a). Model performance was evaluated using $R^2$, and statistical significance was assessed through bootstrap resampling and FDR correction to identify statistically significant activated voxels (see Section A.2 for details).

### 2.2 Linear and Nonlinear Neural Encoding Models

The general neural encoding model can be formulated as follows:

$$\hat{y} = f(x) \tag{1}$$

where $x$ denotes the feature space of external stimuli, $\hat{y}$ is the predicted brain activity space (i.e., BOLD responses in this study), and $f(\cdot)$ denotes the function of neural encoding models. Specifically, the computational representation of visual images spans the feature space, and the preprocessed beta maps serve as the brain activity space ($y$). We constructed both linear and nonlinear neural encoding models in simplified form. Each model consisted of four fully connected layers with skip connections, where the nonlinear neural encoding model additionally incorporated ReLU activation functions between layers (as shown in Fig.2b, and more details in Section A.3).

### 2.3 Computation of the Jacobian Matrix

The Jacobian matrix characterizes the local linear mapping between the representations ($x$) and the predicted neural responses ($\hat{y}$). It reflects the sensitivity of the neural encoding model to variations in the input. Upon completion of model training, we fixed the parameters of the neural encoding model and input the test set data to obtain the corresponding predictions. Given the $i$-th sample $x_i \in \mathbb{R}^D$ in the test set and the prediction of the nonlinear neural neural encoding model $\hat{y}_i = f(x_i) \in \mathbb{R}^M$, the Jacobian matrix $\mathbf{JM}_i$ can be calculated by taking the derivative of the model's output $\hat{y}_i$ with respect to its input $x_i$ as follows:

$$\mathbf{JM}_i = \left(\frac{\partial \hat{y}_i}{\partial x_i}\right)^T = \begin{bmatrix} \frac{\partial \hat{y}_{i,1}}{\partial x_{i,1}} & \cdots & \frac{\partial \hat{y}_{i,M}}{\partial x_{i,1}} \\ \vdots & \ddots & \vdots \\ \frac{\partial \hat{y}_{i,1}}{\partial x_{i,D}} & \cdots & \frac{\partial \hat{y}_{i,M}}{\partial x_{i,D}} \end{bmatrix} \in \mathbb{R}^{D \times M} \tag{2}$$

We denote the collection of Jacobian matrices of all testing samples as $\mathbf{JM} \in \mathbb{R}^{N \times D \times M}$, where $D = 768$ is the dimensionality of the representation in CLIP-ViT, $N = 1000$ is the sample size of the test set, and $M$ is the number of voxels in the brain. Notably, we adopt a forward-mode analytical differentiation approach (Section A.4) to compute Jacobians, accelerating the process by over 99.9% compared to traditional backward-mode automatic differentiation.

### 2.4 Jacobian-based Nonlinearity Evaluation Index

To systematically quantify the nonlinearity characteristics, this study proposed the **J**acobian-based **N**onlinearity **E**valuation index (JNE). This metric established a deviation measure from the linear superposition principle in neural response systems by analyzing inter-sample statistical properties of Jacobian matrices. The computational workflow is defined as follows:

**Step 1: Jacobian matrix centralization**

$$\mathbf{JM}_{mean} = \mathbb{E}[\mathbf{JM}] = \frac{1}{N} \sum_{i=1}^{N} \mathbf{JM}_i \in \mathbb{R}^{D \times M} \tag{3}$$

---

[1] https://huggingface.co/openai/clip-vit-base-patch16

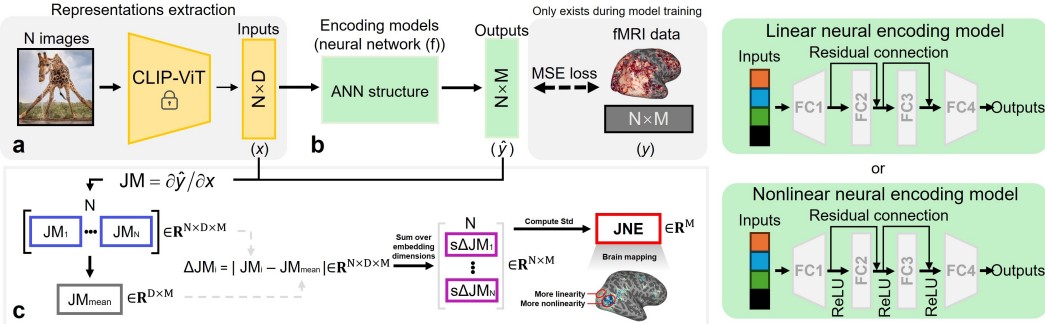

Figure 2: **Framework for representation extraction, neural encoding model, and JNE Computation. a**. Extract ANN representations from images using the CLIP image encoder. **b**. Use the extracted representations in (linear/nonlinear) neural encoding model to predict brain responses to images. **c**. Compute the Jacobian matrix to represent the mapping relationship between inputs and outputs of the neural encoding model. Further, calculate the mean, sum, and standard deviation of the Jacobian matrix to obtain the JNE metric.

This operation constructed a baseline linear mapping matrix by eliminating stimulus-specific idiosyncrasies through sample averaging.

**Step 2: Sample-specific deviation computation**

$$s\Delta\mathbf{JM}_i = \|\mathbf{JM}_i - \mathbf{JM}_{mean}\|_1 \in \mathbb{R}^M \tag{4}$$

A Manhattan norm ($\|\cdot\|_1$) contraction along the feature dimension ($D$-axis) yields an L1 deviation vector in voxel space for each sample. This metric quantified the absolute deviation between local linear approximations (per-sample Jacobians) and the global averaged mapping. Empirical tests in Section A.9.2 show that both L1 and L2 norms yield highly consistent JNE spatial patterns, but we adopt L1 for its robustness in high-dimensional settings [32, 33, 34].

**Step 3: Mean absolute deviation estimation**

$$\Delta\mu = \mathbb{E}[s\Delta\mathbf{JM}] = \frac{1}{N}\sum_{i=1}^{N} s\Delta\mathbf{JM}_i \in \mathbb{R}^M \tag{5}$$

This calculated the cross-sample averaged absolute deviation, establishing a baseline reference for fluctuation intensity of mapping relationships at the voxel level.

**Step 4: Nonlinear dispersion quantification**

$$\text{JNE}_m = \sigma(s\Delta\mathbf{JM}_{\cdot,m}) = \sqrt{\frac{1}{N}\sum_{i=1}^{N}\left(s\Delta\mathbf{JM}_{i,m} - \Delta\mu_m\right)^2} \quad \forall m \in \{1,...,M\} \tag{6}$$

By computing voxel-wise standard deviations ($\sigma(\cdot)$), this precisely characterized the dispersion of mapping relationships across stimuli. This statistic intrinsically reflects second-moment features of Jacobian matrices in sample space, with higher-order nonlinear systems exhibiting significant $\sigma$-value growth. Thus, JNE quantifies nonlinearityvia Jacobian dispersion statistics, approximating voxel-level BOLD nonlinearity. The complete flow of JNE calculation is shown in Fig.2c.

Mathematically, the JNE metric rigorously corresponds to the quantitative evaluation of the homogeneity and superposition principles in linear systems:

• In an ideal linear neural encoding model, where $\mathbf{JM}_i \equiv \mathbf{JM}_{mean}$, $\text{JNE}_m = 0$;

• Nonlinear mechanisms induce sample-dependent Jacobian shifts, with dispersion $\text{JNE}_m$ monotonically increasing with nonlinearity strength.

Critically, both input features and output beta maps were z-score normalized prior to model training and JNE computation. This standardization ensures that the input space, output space, and mapping space operate under consistent scale assumptions, enabling meaningful comparison of Jacobian variability across voxels and brain regions. Therefore, the proposed metric inherently circumvents

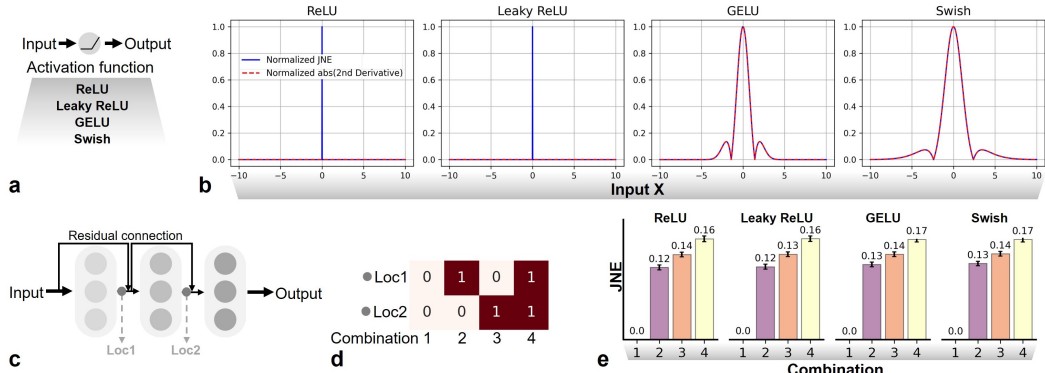

Figure 3: **Validation of the JNE metric in controlled simulation experiments. a**. Simulated neuron model; **b**. Comparison between the normalized JNE (min-max normalization) and the response curves of the normalized second-order derivatives (absolute value and min-max normalization) of various activation functions(Response curves, 1st and 2nd derivatives of the four activation functions in A.5); **c**. Schematic of the ANN-based simulation framework, where the model consists of a three-layer fully connected residual network. Loc1 and Loc2 represent pluggable activation function modules that serve as controllable sources of nonlinearity; **d**. Binary configurations of activation function placement (0 indicates absence, 1 indicates presence); **e**. Comparison of JNE values across four activation functions under different activation placement configurations (mean±std).

cross-regional scale normalization—the magnitude of standard deviation directly encodes biological sensitivity differences in neural responses. Stimulus-sensitive brain regions (e.g., early visual cortex) naturally exhibit greater Jacobian variability across inputs, which mathematically aligns with the sample-dependent characteristics of nonlinear mechanisms. This approach preserves neurophysiological interpretability while providing a unified measurement benchmark for cross-regional nonlinearity comparisons.

## 3 Results

### 3.1 Simulation-Based Validation of JNE

**Single-neuron-level validation of activation functions.** To validate the ability of JNE to accurately characterize system-level nonlinearity, we first constructed a simple and controllable simulation framework. Specifically, we focused on four commonly used activation functions—ReLU, Leaky ReLU, GELU, and Swish—and designed a minimal neuron model consisting solely of input, nonlinear transformation, and output (Fig.3a). JNE essentially measures the "statistical dispersion" of the local linear mappings (i.e., Jacobians) of a neuron's response to input perturbations across samples. Given an activation function $\phi(x)$, its first derivative $\phi'(x)$ corresponds to the Jacobian matrix. If $\phi'(x)$ varies significantly within a local neighborhood $[x - \delta, x + \delta]$, it indicates that the neuron's response exhibits non-uniform shifts with input changes—thus violating linear assumptions. When $\delta \to 0$, this rate of change can be described by the second derivative $\phi''(x)$. Therefore, in theory, the second derivative of $\phi(x)$ can serve as a **structural reference** for JNE: a larger $\phi''(x)$ implies stronger local nonlinearity, which should be reflected by a higher JNE value. In our experiment, inputs were defined over the range $[-10, 10]$, and a sliding window $[x - \delta, x + \delta]$ (with $\delta = 0.001$ and a step size of 0.02) was used. For each central point $x$, we sampled 1001 perturbed inputs within the local window and computed the sequence of first derivatives of the activation function to calculate the JNE value. This process was repeated over the entire input range to produce JNE curves, which were then compared to the second derivative curves of the corresponding activation functions.

As shown in Fig.3b, the JNE curves closely resemble the structural patterns of the second derivatives for all activation functions. Smooth functions like GELU and Swish exhibited high JNE values over broad input ranges, whereas ReLU only produced a peak around its discontinuity at $x = 0$, with a value of zero elsewhere. Additionally, JNE was strictly zero in linear regions of the activation functions, validating its theoretical soundness. However, it is worth noting that JNE measures the dispersion of local linear mappings across input samples, which correlates with the magnitude of second derivatives in theory, but is not identical due to its statistical nature.

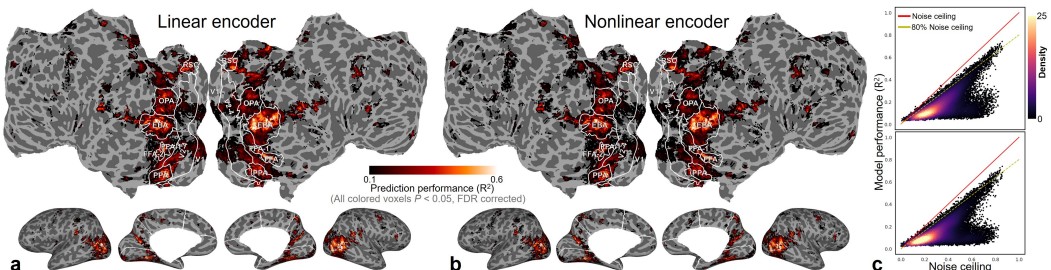

Figure 4: **Comparison of prediction performance between linear and nonlinear neural encoding model. a**. Prediction performance ($R^2$) of the linear neural encoding model. **b**. $R^2$ of the nonlinear encoder. **c**. 2D histogram of linear and nonlinear neural encoding model performance relative to the noise ceiling (calculation in [30]) of significantly activated voxels and the 80% noise ceiling. For visualization purposes, only voxels with prediction performance significantly higher than chance ($P<0.05$, FDR corrected, one-sided t-test) are displayed in the brain maps.

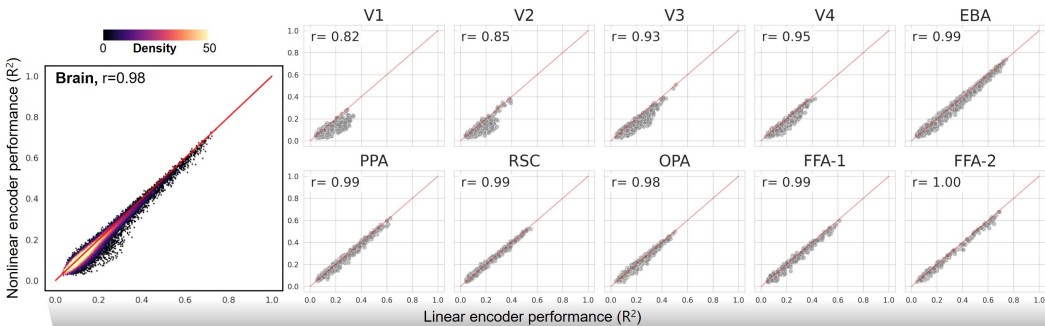

Figure 5: **Scatter plot comparison between linear and nonlinear neural encoding models.** Scatter plots comparing the $R^2$ of linear and nonlinear neural encoding models across the whole brain and 10 regions of interest (ROIs) defined by the HCP-MMP atlas [35]. The Pearson correlation coefficient between the $R^2$ sequences of the two models is also reported.

**Network-level validation via ANN-based simulation.** To further assess JNE's practical utility in multilayer network structures, we developed an ANN simulation framework (Fig.3c). The model consisted of a three-layer fully connected residual architecture and was not subject to any training—it was used solely for forward propagation to obtain the Jacobian matrices between inputs and outputs. Nonlinearity was introduced via two plug-in locations (Loc1 and Loc2), where activation functions could be flexibly inserted in a fully controlled manner (Fig.3c and d). This setup allowed us to systematically evaluate JNE's sensitivity to different nonlinear configurations without altering the overall network architecture. To simulate the real neural encoding models used in this study, we set the input dimension to 768 (mimicking CLIP-ViT image representations) and generated an input matrix of size $1000 \times 768$, with the corresponding output being a simulated response vector of size $1000 \times 1$. For each input sample, we computed the Jacobian of the output with respect to the input and evaluated the overall nonlinearity using the standard JNE pipeline. We conducted comparative analyses for four activation functions under different Loc1/Loc2 configurations (none/shallow/deep/both layers). Each configuration was sampled 200 times to ensure stable statistical estimates (see Section A.6 for details).

As illustrated in Fig.3e, the JNE value was zero under the ideal linear system (no activation function), consistent with theoretical expectations. Upon the introduction of activation functions, JNE values increased significantly. Interestingly, we observed that the position of the nonlinearity affected the results: configurations with activations in the deeper layer (Loc2) generally yielded higher JNE values compared to those in the shallow layer (Loc1), and the two-layer configuration led to the strongest nonlinearity. This trend was consistent across all activation functions, revealing an activation-invariant pattern.

These results demonstrate that JNE not only reconstructs the intrinsic nonlinear structures of activation functions at the theoretical level but also reliably detects and quantifies diverse sources of nonlinearity

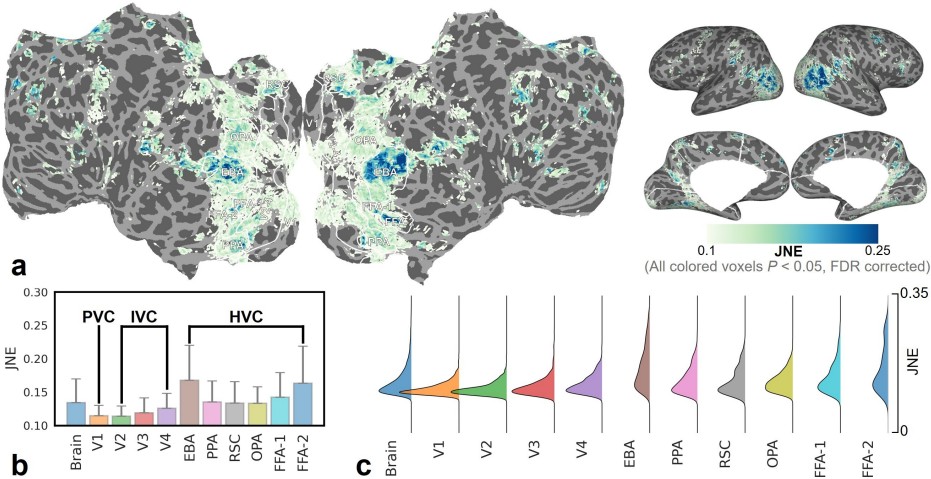

Figure 6: **Nonlinear distribution in the brain. a**. Activation map of JNE across the brain, illustrating the spatial distribution of model-predicted BOLD response nonlinearity. **b**. Mean JNE values across the whole brain and 10 visual ROIs (mean±std). **c**. JNE distribution across the whole brain and 10 visual ROIs, with the x-axis indicating density.

in complex networks—validating its effectiveness as a nonlinearity metric, paving the way for approximating BOLD nonlinearity in fMRI.

## 3.2 Nonlinearity Analysis Based on Neural Encoding Models Comparison

As a response to the introduction, this paper begins by addressing a central question: Can we infer the nonlinearity of individual voxels by comparing the $R^2$ of linear versus nonlinear neural encoding models? As a first step, we extracted embedding features from the final layer of CLIP-ViT and aligned them to fMRI responses using both a linear and a nonlinear neural encoding model, respectively. Fig.4 presents the relevant results for Subject 1 (all results in the main text refer to Subject 1; other subjects in Section B).

As shown in Figs.4a and 4b, both types of models exhibit highly consistent predictive performance across the whole brain, with activations primarily concentrating in the primary, intermediate, and higher-order visual cortices, as well as parts of the prefrontal cortex. Furthermore, we observed that both models achieved predictive performance close to the noise ceiling for many voxels (Fig.4c), with some voxels even exceeding 80% of the noise ceiling. This indicates that both linear and nonlinear neural encoding models can effectively account for voxel responses in the held-out test data (explaining up to 74% of the response variance). To further assess the consistency of the two models in predictive performance, we conducted a correlation analysis between the $R^2$ sequences of the linear and nonlinear neural encoding models across the whole brain and within ten predefined vision-related ROIs (Fig.5). Results show strong positive correlations between the $R^2$ sequences of the two models in all ROIs, with the lowest Pearson correlation observed in V1 (min $r = 0.82$), indicating a strong relationship between the activation patterns of the two neural encoding models.

Together, these findings support the notion that when the deep features of a visual encoder have already sufficiently captured the nonlinear transformations of visual information, a linear neural encoding model can still effectively fit neural response patterns (Fig.1b, demonstrating the high similarity between linear and nonlinear neural encoding models). In other words, the difference in predictive performance between linear and nonlinear neural encoding models becomes marginal, making it unreliable to infer nonlinearity solely based on model performance comparisons. To address this limitation, we proposed JNE, a concise and principled metric for characterizing nonlinearity. In the following section, we apply JNE to fMRI data as real-world validation.

## 3.3 Spatial Distribution of BOLD Response Nonlinearity in Visual Cortex

Fig.6a illustrates the spatial distribution of JNE across the cortical surface. We observed that the nonlinearity of neural responses was not uniformly distributed across the brain but was instead

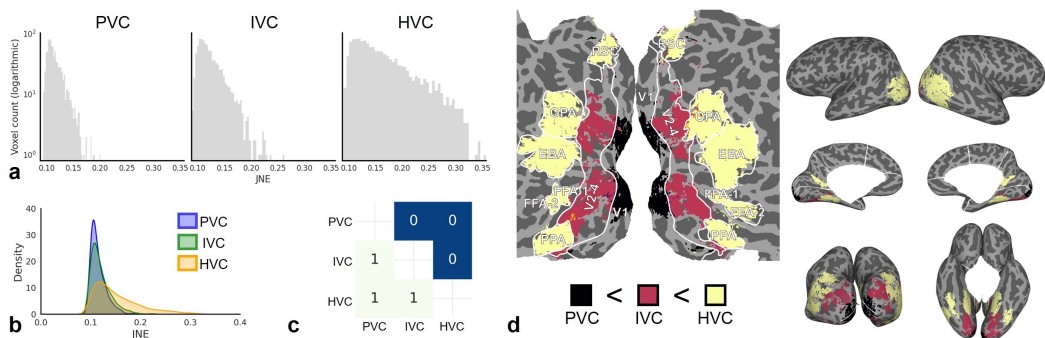

Figure 7: **Hierarchical analysis of nonlinear characteristics in visual cortex. a**. Bar plot comparison of mean JNE values across primary, intermediate, and higher-order visual cortices; **b**. Distributional differences in JNE across the three cortical levels; **c**. Significance matrix of pairwise JNE differences among primary, intermediate, and higher-order visual cortices (one-sided test, $P < 0.05$), where entry $(i, j) = 1$ indicates that cortex $i$ exhibits significantly higher JNE than cortex $j$; **d**. Region-level visualization of hierarchical nonlinearity based on the significance matrix in **c**. Colors represent cortical levels (primary = black, intermediate = red, higher-order = yellow), not voxel-wise specificity. This visualization validates the consistent hierarchical trend across the visual cortex.

concentrated in higher-order cortical regions—including higher-order visual cortices, the temporo-parietal-occipital junction (TPOJ), and parts of the prefrontal cortex. Figs.6b-c and Fig.17 further quantify the statistical characteristics of JNE across the whole brain and within ten predefined ROIs. The **P**rimary **V**isual **C**ortex (PVC-V1) exhibited the lowest mean JNE values, with a distribution tightly concentrated in the low-value range, suggesting that neural responses in this region are more consistent with linear characteristics. In **I**ntermediate **V**isual **C**ortex (IVC-V2, V3, V4), JNE values showed a moderate increase and broader spread, indicating the presence of moderate nonlinear response components. In contrast, **H**igher-order **V**isual **C**ortex (HVC-EBA, PPA, RSC, OPA, FFA-1 and FFA-2) demonstrated the highest JNE values, with distributions skewed toward higher values. This pattern suggests that these regions exhibit more pronounced nonlinear response properties, approximating hierarchical BOLD patterns. Collectively, these results indicate that brain regions exhibiting stronger JNE values, approximating hierarchical BOLD nonlinearity in higher-order cortices, which are typically involved in advanced information integration and abstract semantic processing.

### 3.4 Hierarchical Progression of Nonlinearity Across the Visual Cortex

Furthermore, we observed a hierarchical progression of JNE-approximated BOLD nonlinearity across the visual cortex. As shown in Fig.7a, bar plots comparing JNE across primary, intermediate, and higher-order visual cortices reveal a clear trend: the number of voxels exhibiting high JNE values increases with cortical hierarchy. Fig.7b further illustrates the distribution of JNE values within each visual cortical tier. In PVC, JNE values are predominantly concentrated in the lower range. In IVC, the distribution broadens with a slight shift toward higher values. In contrast, HVC displays a significantly right-skewed distribution with a pronounced long-tail pattern, indicating stronger nonlinear response characteristics. Figs.7c and 7d present heatmaps and statistical comparisons that highlight the significant differences in JNE across the three cortical levels. These results reveal a robust, layer-wise enhancement of nonlinearity across the visual hierarchy—i.e., primary < intermediate < higher-order visual cortex. These findings suggest that JNE, as applied to the neural encoding model, uncovers a progressive nonlinear hierarchy in the predicted BOLD responses across the visual system, beginning in the primary cortex, transitioning through intermediate areas, and peaking in higher-level regions (Fig.7d). This hierarchical organization of BOLD response nonlinearity aligns well with previous neurophysiological evidence supporting functional stratification across the visual cortex [36, 37, 38].

### 3.5 Sample Selectivity of BOLD Response Nonlinearity

To characterize the sample-specific nonlinear responses of different brain regions to visual stimuli, we further introduced the **JNE** with **S**ample-**S**pecificity (JNE-SS$\in \mathbb{R}^{N \times M}$, theoretical details in Section A.10). In brief, JNE-SS quantifies the degree of nonlinearity in a voxel's response to individual image

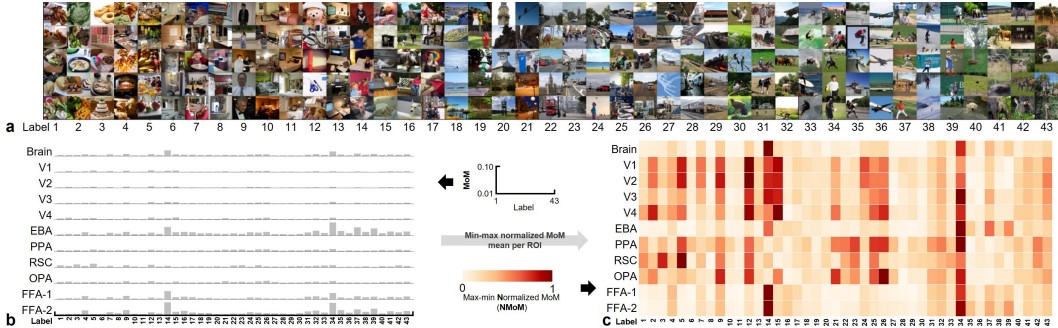

Figure 8: **Sample-specific analysis of brain nonlinear encoding. a**. The test set of 1,000 images was divided into 43 categories and sorted by category; the top 5 images in each category were selected for display (see detailed results in Fig.12). **b**. MoM was computed across the whole brain and within 10 predefined ROIs. **c**. The results in (**b**) were normalized within each ROI using min-max normalization to reflect category preferences of each ROI. Note that the heatmap in Fig. 8c applies min-max normalization within each ROI, which may attenuate visual differences in absolute MoM values. For quantitative comparisons across regions, we refer to Table 5 in Section A.12, which reports MoM values without applying min–max normalization.

stimuli: a lower JNE-SS value for a voxel-image pair indicates that the voxel exhibits a more linear response preference for that specific stimulus, whereas a higher value implies stronger nonlinear tuning. For practical analysis, we first applied t-SNE for 2D embedding of all image stimuli and then performed K-means clustering to categorize them into 43 clusters. This clustering structure preserved a meaningful continuum along the dimension of semantic complexity (Fig. 8a; more details in Section A.11). Subsequently, we computed the **M**ean **of J**NE-SS (MoJ $\in \mathbb{R}^{43 \times M}$) for each voxel across image clusters, and further derived the **M**ean **of M**oJ (MoM $\in \mathbb{R}^{43 \times 11}$) at the whole-brain and ROI levels (including the whole brain and 10 ROIs). This metric reflects the variation in BOLD response nonlinearity elicited by different image categories across brain regions.

The result reveals that MoM values are generally lower in primary and intermediate visual cortices, while notably higher in higher-order visual areas—particularly the EBA (Fig. 8b, Fig. 21a), corroborating our previous findings in Fig. 6. Additionally, EBA maintains consistently high JNE-SS values across multiple image categories, suggesting a potentially central role of EBA in mediating nonlinear visual processing. Further examination of regional nonlinearity preferences for specific image types revealed additional insights (Fig. 8c, Fig. 21b). Images depicting outdoor scenes or complex backgrounds evoked stronger nonlinear responses in PPA and RSC, while stimuli involving people or faces induced pronounced nonlinear fluctuations in FFA and EBA. These findings align with the known functional specificity of these regions [2, 30, 39, 40, 41]. Interestingly, we also observed broad nonlinear responses in primary and intermediate visual areas across a wide range of image categories (Fig.21b), contrasting with the more selective tuning seen in higher-order visual regions.

## 4    Conclusion and Discussion

In this work, we introduce JNE, a novel interpretability metric for nonlinear models via Jacobian dispersion, approximating voxel BOLD nonlinearity. Through rigorous simulation experiments and its application to fMRI data from natural scene viewing, we validate JNE's effectiveness in discerning hierarchical gradients and stimulus-selective patterns of nonlinearity across the visual cortex. Importantly, JNE evaluates input-dependent variability in the model's local linear mappings to BOLD signals, serving as a diagnostic tool for encoding model interpretability rather than a direct assay of intrinsic neural nonlinearity.

The metric deliberately retains scale information in its Jacobian-based standard deviation, as variations in predicted response amplitudes may encode physiologically relevant differences in neural gain. This preserves neuroscientific utility, despite the inherent coupling of gain and nonlinearity. Although JNE could in principle confound response gain, nonlinearity degree, and input dimensionality, theoretical derivations (Section A.9.1) and empirical assessments on real data (Section A.9.2) demonstrate that

these confounds are minimal under the constrained architectures and stimulus conditions typical of neural encoding frameworks.

In real data experiments, our study demonstrates that the nonlinear characteristics of neural responses exhibit significant differences across brain regions (Section 3.3). Specifically, JNE values are generally lower in primary cortices and significantly higher in higher-order cortices. The elevated JNE values in higher-order regions suggest that the encoding model must employ more nonlinear mappings to predict BOLD responses in these areas, which is consistent with the hypothesis that these areas engage in more nonlinear processing of visual information. However, JNE captures the net nonlinearity of the input-to-BOLD pathway, which integrates both neural and hemodynamic factors.

In addition, we observed a hierarchical trend of BOLD response nonlinearity within the visual cortex (Section 3.4). This result reveals a gradient pattern consistent with the hierarchical organization of the visual processing pathways [36, 37, 38]. Specifically, PSV (V1) is primarily responsible for processing low-level perceptual features such as edges, orientation, and spatial frequency [30, 35]. The low JNE values in this region indicate that it may perform only simple linear processing of visual information. Subsequently, the processed information is transmitted along the dorsal and ventral visual pathways to IVC and HVC, during which JNE values progressively increase. The observed monotonic increase in JNE from primary to higher-order visual cortices supports a hierarchical progression of information processing along the ventral visual stream, consistent with established models of visual cortical organization. [2, 35, 42, 43, 44]. We note that fMRI responses integrate both feedforward and feedback signals, and thus JNE captures the net nonlinearity of the system under natural viewing conditions. Additionally, it suggests that the propagation of visual information in the brain may follow a progressive pattern from low-nonlinearity (low JNE) regions to high-nonlinearity (high JNE) regions, gradually transitioning from linear to nonlinear processing.

Section 3.5 further supports this view, as reflected in the relatively low MoM values in PVC, contrasted with the higher MoM values in higher visual cortices (such as PPA, RSC, FFA), which also exhibit stronger selectivity to different image categories. Moreover, we found that PPA and RSC showed stronger nonlinear responses to outdoor scenes, whereas FFA displayed highly significant nonlinear responses to faces and human figures. These results corroborate the functional selectivity of these regions to different stimuli [2, 30, 45], and also suggest that the higher nonlinearity observed in these areas may underlie their roles in supporting semantic generalization, contextual integration, and complex perceptual discrimination—core aspects of high-level visual cognition. Notably, we also found that even in V1–V4, MoM values increased when images contained rich backgrounds, multiple objects, or higher visual complexity, revealing a limited yet genuine capacity for nonlinear modulation in early visual processing stages.

Generally, the primary contribution of this work is the proposal and validation of the JNE metric. While we demonstrate its application in revealing nonlinearity in visual cortex, these results should be seen as preliminary validations of the method's feasibility.

## 5  Limitations and Prospects

Despite its contributions, this study has several limitations that suggest avenues for future work. First, JNE approximately quantifies nonlinearity in mappings to BOLD signals rather than directly to neural activity. BOLD is an indirect hemodynamic measure confounded by vascular factors (e.g., blood flow refractory effects and HRF nonlinearity), which cannot be fully separated from neural signals in standard fMRI. Thus, JNE reflects the composite nonlinearity of the input-to-BOLD pathway, limiting direct neuronal inferences. While simulations used a controllable ANN framework, we did not exhaustively assess how activation function count, placement, or type influences overall nonlinearity, nor track its evolution during training. Furthermore, the computation of JNE essentially depends on the specific encoding model architecture and stimulus distribution used. Future studies should systematically evaluate the consistency of JNE across different model architectures (e.g., CNNs, Transformers), training objectives (supervised, self-supervised, unsupervised), and stimulus sets, thereby validating its robustness as a comparative tool. Additionally, no absolute JNE threshold was defined to classify voxels as "linear" or "nonlinear," restricting analyses to relative comparisons and voxel-level conclusions. Moreover, our sample-specific analysis relied on informal t-SNE ordering for image categories, lacking rigorous standards. Finer-grained feature/patch-level investigations are needed to pinpoint what drives region-specific nonlinear responses.

# 6    Acknowledgements

This work was supported by the National Natural Science Foundation of China under Grants 62076205 and 62576065.

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

# A Appendix A

## A.1 Theoretical Basis of Nonlinearity Measurement

In a finite-dimensional space, an ideal linear system (here the system refers to a neural encoding model) should satisfy two fundamental principles: additivity ($f(x_1 + x_2) = f(x_1) + f(x_2)$) and homogeneity ($f(ax) = af(x)$) [46, 47]. These two properties must hold simultaneously for the system to be formalized as $y = Wx$, where $W$ is a mapping matrix that remains invariant across samples. A nonlinear system no longer adheres to this constant mapping. Its input-output relation $f(x_i)$ can instead be locally approximated as $y_i = W_i x_i$, where $W_i$ denotes the local derivative (i.e., the Jacobian matrix) corresponding to the $i$-th input sample. Since the system's response to input perturbations depends on the current input, $W_i$ varies across samples, thereby violating the input-invariance property of a linear system.

Based on this analysis, the present study constructs a quantitative measure of deviation from linearity by evaluating the statistical variation of Jacobian matrices across samples. Specifically, if $W_i$ remains consistent across samples, the system can be regarded as linear; conversely, greater variability of $W_i$ indicates a stronger deviation from linearity and hence a higher degree of nonlinearity in the neural response at that region or voxel.

## A.2 fMRI Data Description, Preprocessing, and Statistical Significance Evaluation

This study uses the Natural Scenes Dataset (NSD) [30], which contains fMRI recordings from eight subjects passively viewing 73,000 colored natural images over a period of more than 40 hours. The images are sourced from the MS-COCO dataset [48], with each image presented for 3 seconds and repeated three times across 30–40 scanning sessions. Each subject completed approximately 22,000 to 30,000 trials. The fMRI data were acquired using a whole-brain 7T gradient-echo EPI sequence with a spatial resolution of 1.8 mm and a TR of 1.6 seconds. Single-trial beta maps were estimated using a customized general linear model (GLM) and released alongside the raw fMRI data [49]. Following previous studies [49], the beta maps were normalized (zero mean and unit variance) within each scan run and averaged across repeated presentations of the same image, serving as functional measures of brain activity.

In our experiments, each image and its corresponding averaged beta map are considered a sample, with an 8:1:1 split into training, validation, and test sets (8000:1000:1000 samples). The training and validation sets involve non-overlapping subjects, while the test set includes repeated samples across subjects. Notably, S1, S2, S5, and S7 completed the full experimental protocol, and their fMRI data are used for analysis in this study.

We use the pre-trained CLIP-ViT model [31] to extract deep visual representations of the stimuli and construct both linear and nonlinear neural encoding models to predict fMRI responses.

In addition, we use the coefficient of determination ($R^2$) to assess model performance on the test set. To determine the statistical significance of the predictions, we adopt the approach described in [2], performing 200 bootstrap resampling iterations on the test data and computing FDR-corrected $P$-value thresholds for multiple evaluation metrics [2, 50]. As a result, all activated voxels reported in this study exhibit statistically significant activation after FDR correction, confirming task-specific neural responses under rigorous statistical criteria.

## A.3 Training Strategy

The mean squared error (MSE) [51] is used as the loss function in the encoder, and the training process of the encoder utilizes the Adam optimizer [52] for parameter optimization. To prevent overfitting, early stopping is employed, ceasing training if the validation loss fails to improve over 8 consecutive epochs. Additionally, the model parameters that demonstrate the optimal performance on the validation set are recorded and preserved.

## A.4 Efficient Jacobian Computation via Forward-Mode Analytical Differentiation

To address computational costs, traditional backward computation methods (e.g., using *torch.autograd.grad* in PyTorch with a pretrained model and corresponding inputs/outputs) require

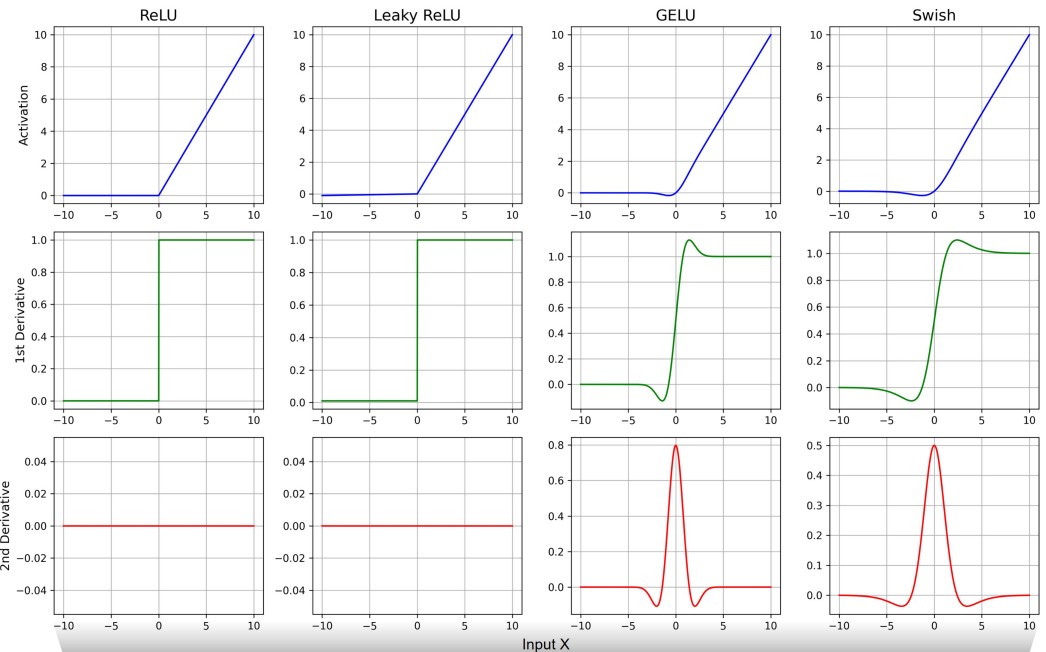

Figure 9: **Visualization of activation function response and derivative characteristics.** We present the response curves (first row), first-order derivative curves (second row), and second-order derivative curves (third row) of four activation functions—ReLU, Leaky ReLU, GELU, and Swish—over the input range $[-10, 10]$.

approximately several days (Intel Xeon E5-2620 v4 CPU in this study). We optimized this process through explicit forward-mode analytical differentiation. For the nonlinear model employed in our study, the forward structure is as follows, where $h^{(k)}$ denotes the hidden activations at layer $k$, $b_k$ represents the bias vectors, $W_{FCk}$ are the weight matrices, and $\phi_1, \phi_2, \phi_3$ are the activation functions:

$$
\begin{aligned}
h^{(0)} &= \phi_1(W_{FC1}x_i + b_1), \\
h^{(1)} &= \phi_2(W_{FC2}h^{(0)} + b_2) + h^{(0)}, \\
h^{(2)} &= \phi_3(W_{FC3}h^{(1)} + b_3) + h^{(1)}, \\
\hat{y}_i &= W_{FC4}h^{(2)} + b_4.
\end{aligned}
\tag{7}
$$

Since commonly used activation functions (e.g., ReLU, GELU) possess analytical first-order derivatives, we record the sample-specific derivative matrices $W_{\phi_k,i}$ during forward propagation and leverage them to compute the full Jacobian:

$$
\begin{aligned}
J_i = {} & W_{FC1}W_{\phi_1,i}W_{FC2}W_{\phi_2,i}W_{FC3}W_{\phi_3,i}W_{FC4} + W_{FC1}W_{\phi_1,i}W_{FC3}W_{\phi_3,i}W_{FC4} \\
& + W_{FC1}W_{\phi_1,i}W_{FC2}W_{\phi_2,i}W_{FC4} + W_{FC1}W_{\phi_1,i}W_{FC4}.
\end{aligned}
\tag{8}
$$

This approach reduces the computation time to 20–40 seconds, achieving an acceleration of over 99.9%.

### A.5 Response Curves and Derivative Characteristics of Activation Functions

This section presents the response curves, first-order derivatives, and second-order derivatives of four commonly used activation functions—ReLU, Leaky ReLU, GELU, and Swish—over the input range $x \in [-10, 10]$. By visualizing their local responses and higher-order derivative characteristics, these plots provide intuitive references for analyzing nonlinear structures (Fig.9).

### A.6 Details of the Simulation Procedure

To ensure that the evaluation of JNE in the simulation environment is both statistically robust and structurally controlled, we designed the following experimental procedure:

**Step1-Model architecture setup.** We constructed a three-layer feedforward artificial neural network (ANN) (see Fig. 3c), where all layers are fully connected. The input and hidden layers are both set to 768 dimensions, and the output layer has a single unit, simulating voxel-level prediction in neural encoding models. All network weights are randomly initialized prior to each experimental run to ensure generalizability and representativeness of the results.

**Step2-Activation function insertion strategy.** To introduce nonlinearity, activation functions are inserted between network layers. Four configurations of activation function placement are constructed (see Fig. 3d), ranging from a fully linear structure (no activation functions) to configurations where activations are applied between all layers. Identical weights are maintained across all configurations to ensure fair comparison.

**Step3-Input data generation.** The input feature matrix is defined as $\mathbb{R}^{1000 \times 768}$, with each row representing a high-dimensional feature vector of a simulated sample. All elements are sampled from a uniform distribution within the fixed interval $[-10, 10]$, ensuring sufficient randomness and variability in the input space.

**Step4-Jacobian matrix and JNE computation.** For each activation function configuration, the Jacobian matrix of the ANN is computed for each sample using automatic differentiation tools implemented in Python. The Jacobian has a shape of $1000 \times 768 \times 1$, and is used to calculate the JNE metric. One JNE value is obtained per configuration, resulting in a total of four values.

**Step5-Repeated experiments for statistical robustness.** To enhance statistical reliability, Steps 1 through 4 are repeated 200 times with independently initialized weights. This results in a $200 \times 4$ JNE response matrix, allowing for a systematic characterization of how nonlinearity varies with different activation placements in the ANN.

**Step6-Comparison of activation function types.** To comprehensively evaluate the nonlinear characteristics of different activation functions, we introduce four commonly used types: ReLU, Leaky ReLU, GELU, and Swish. For each activation function, Steps 1 through 5 are repeated, yielding a complete JNE response matrix with dimensions $4 \times 200 \times 4$ (4 activation types $\times$ 200 repetitions $\times$ 4 placement configurations), which enables detailed analysis of activation-specific trends and distinctions under the JNE metric.

## A.7 Robustness to Input Sample Size, Distribution and Scale

To address the sensitivity of the JNE metric to the choice of input set and sample size, we conducted controlled simulation experiments. These simulations systematically evaluated the stability and robustness of JNE under variations in sample size, input perturbations, distribution types, and numerical scales. The results indicate that JNE exhibits strong robustness to input perturbations and remains stable and reliable even with relatively small sample sizes.

### A.7.1 Experiment on Sample Size Variation

The estimation of JNE can be influenced by sample size and input perturbations. To quantify this impact and assess whether moderate sample sizes suffice for stable performance, we conducted a simulation evaluating JNE's ability to track a known modulation pattern under varying $N$. The details of simulation experiments are as follows:

- **Sample size variation:** $N \in \{10, 20, 30, 50, 100, 200, 300, 500, 1000\}$.
- **Signal perturbation factors:** We generated 1001 evenly spaced points in the interval $x \in [0, 2\pi]$, defined a modulating signal $c = \sin(x)$, and used $|c|$ as the input perturbation factor.
- **Jacobian simulation:** For each $N$, we generated random tensors $J \in \mathbb{R}^{N \times 768 \times 1} \sim \mathcal{N}(0, 1)$, then modulated them pointwise using $|c|$ to obtain 1001 Jacobian sequences under varying input amplitudes.
- **JNE computation:** We computed JNE values for each modulated Jacobian sequence using the standard pipeline.
- **Evaluation metric:** We measured the Pearson correlation coefficient $r$ between the resulting JNE curve and the target trend $|\sin(x)|$ to assess tracking accuracy.

Generally, from Table 1, we find that even with $N = 10$, the estimated JNE shows a strong correlation with the target modulation ($r = 0.87$). Performance improves with increasing $N$, achieving near-perfect alignment ($r \geq 0.99$) for $N \geq 100$. These findings indicate that JNE is robust to input perturbations and stabilizes effectively with reasonable sample sizes.

Table 1: Correlation coefficients ($r$) between JNE and $|\sin(x)|$ across sample sizes.

| Sample size $N$ | 10 | 20 | 30 | 50 | 100 | 200 | 300 | 500 | 1000 |
|---|---|---|---|---|---|---|---|---|---|
| Correlation $r$ | 0.87 | 0.94 | 0.96 | 0.97 | 0.99 | 1.00 | 1.00 | 1.00 | 1.00 |

### A.7.2 Experiment on Input Distribution and Scale

Evaluating sensitivity to input distributions is essential for verifying JNE's stability and generalizability. We systematically varied distribution type and scale while maintaining the network architecture from *Network-level validation via ANN-based simulation* in Section 3.1. Generally, the simulation experiments are defined as follows:

- **Distribution type:** Inputs were sampled from either a uniform distribution $\mathcal{U}(a, b)$ or a normal distribution $\mathcal{N}(\mu, \sigma^2)$.
- **Numerical scale:** Three amplitude ranges were tested—small $[-100, 100]$, medium $[-1000, 1000]$, and large $[-10000, 10000]$.
- **Network configurations:** Under each input setting, we evaluated JNE across four activation functions (ReLU, Leaky ReLU, GELU, Swish) and four activation location settings (Loc Combination $\in \{1, 2, 3, 4\}$, see Fig. 3d for details).
- **Input size and output size:** Input fixed at $1000 \times 768$, with output as $1000 \times 1$. Each configuration was sampled 200 times for statistical estimates.

Table 2: Mean JNE values under uniform distributions with varying scales.

| Distribution $\mathcal{U}(a, b)$ | $a = -100, b = 100$ | | | | $a = -1000, b = 1000$ | | | | $a = -10000, b = 10000$ | | | |
|---|---|---|---|---|---|---|---|---|---|---|---|---|
| | Loc 1 | Loc 2 | Loc 3 | Loc 4 | Loc 1 | Loc 2 | Loc 3 | Loc 4 | Loc 1 | Loc 2 | Loc 3 | Loc 4 |
| ReLU | 0.00 | 0.12 | 0.14 | 0.16 | 0.00 | 0.12 | 0.13 | 0.16 | 0.00 | 0.12 | 0.14 | 0.16 |
| Leaky ReLU | 0.00 | 0.11 | 0.13 | 0.16 | 0.00 | 0.12 | 0.14 | 0.16 | 0.00 | 0.11 | 0.13 | 0.16 |
| GELU | 0.00 | 0.12 | 0.14 | 0.16 | 0.00 | 0.12 | 0.14 | 0.16 | 0.00 | 0.12 | 0.14 | 0.16 |
| Swish | 0.00 | 0.12 | 0.14 | 0.16 | 0.00 | 0.12 | 0.14 | 0.16 | 0.00 | 0.12 | 0.14 | 0.16 |

**Mean JNE Values under Uniform Input Distributions.** Results are highly consistent across scales, indicating insensitivity to input amplitude changes (see Table 2).

Table 3: Mean JNE values under normal distributions with varying variances.

| Distribution $\mathcal{N}(\mu, \sigma^2)$ | $\mu = 0, \sigma^2 = 100$ | | | | $\mu = 0, \sigma^2 = 1000$ | | | | $\mu = 0, \sigma^2 = 10000$ | | | |
|---|---|---|---|---|---|---|---|---|---|---|---|---|
| | Loc 1 | Loc 2 | Loc 3 | Loc 4 | Loc 1 | Loc 2 | Loc 3 | Loc 4 | Loc 1 | Loc 2 | Loc 3 | Loc 4 |
| ReLU | 0.00 | 0.12 | 0.14 | 0.16 | 0.00 | 0.12 | 0.14 | 0.16 | 0.00 | 0.12 | 0.14 | 0.16 |
| Leaky ReLU | 0.00 | 0.12 | 0.13 | 0.16 | 0.00 | 0.11 | 0.13 | 0.16 | 0.00 | 0.11 | 0.13 | 0.16 |
| GELU | 0.00 | 0.12 | 0.14 | 0.16 | 0.00 | 0.12 | 0.14 | 0.16 | 0.00 | 0.12 | 0.14 | 0.16 |
| Swish | 0.00 | 0.12 | 0.14 | 0.16 | 0.00 | 0.12 | 0.14 | 0.16 | 0.00 | 0.12 | 0.14 | 0.16 |

**Mean JNE Values under Normal Input Distributions.** Under varying variance configurations of $\mathcal{N}(\mu, \sigma^2)$, JNE remains nearly unchanged, confirming robustness to distribution types (see Table 3).

### A.8 Theoretical Justification of JNE as a Proxy for BOLD Response Nonlinearity

The JNE metric quantifies a model's sensitivity to input perturbations, defined as the standard deviation of the Jacobian norm computed across different input samples (i.e., predicted fMRI responses). By the chain rule, the complete input–output mapping in a neural encoding framework can be expressed as:

$$I \xrightarrow{m(\cdot)} x \xrightarrow{f(\cdot)} \hat{y} \tag{9}$$

Here, $I$ denotes the image stimuli, $m(\cdot)$ denotes a pretrained ANN that extracts high-level features $x$ from the input $I$, and $f(\cdot)$ is a linear or nonlinear neural encoding model that maps the features into the BOLD response space $\hat{y}$. Therefore, applying the chain rule gives:

$$\frac{\partial \hat{y}}{\partial I} = \frac{\partial \hat{y}}{\partial x} \cdot \frac{\partial x}{\partial I} = \frac{\partial f(x)}{\partial x} \cdot \frac{\partial x}{\partial I} \tag{10}$$

Since the feature extractor $m(\cdot)$ is fixed, the term $\partial x / \partial I$ is constant across all voxels. Therefore, voxel-wise differences in input–output sensitivity are entirely determined by $\partial f(x)/\partial x$. From a physiological perspective, the actual brain response pathway can be approximated as:

$$I \xrightarrow{g(\cdot)} \text{Neural Activity} \xrightarrow{z(\cdot)} y \tag{11}$$

where $g(\cdot)$ denotes the human brain system, $z(\cdot)$ denotes the fMRI signal acquisition equipment, and $y$ denotes the BOLD responses. If the ANN representation $x = m(I)$ partially approximates the neural state (a widely supported assumption in prior neural encoding work), then $\partial f(x)/\partial x$ can be viewed as a first-order approximation of $\partial z(g(I))/\partial I$. That is:

$$\frac{\partial f(m(I))}{\partial I} \propto \frac{\partial f(x)}{\partial x} = \frac{\partial \hat{y}}{\partial x} \propto \frac{\partial z(g(I))}{\partial I} = \frac{\partial y}{\partial I} \tag{12}$$

Thus, variation in the JNE score reflects the degree of nonlinear sensitivity in the composite system $z(g(\cdot))$. JNE only increases when this system exhibits substantial input-dependent nonlinearity, providing theoretical justification for interpreting JNE as an approximate proxy for BOLD response nonlinearity.

### A.9  Theoretical and Empirical Analysis of JNE Sensitivity

As described in Section 2.4, the JNE metric is designed to quantify nonlinearity in neural encoding models by measuring the dispersion of sample-specific Jacobian deviations from their global mean. While this approach provides a principled interpretability tool, it inherently incorporates elements of response gain (magnitude of neural responses) alongside structural nonlinearity and dependencies across input dimensions. This integration is intentional, as it allows JNE to reflect physiologically relevant sensitivity differences across voxels or brain regions under standardized input and output spaces (e.g., z-scored features and beta maps). However, this design choice raises potential concerns about interpretability, such as whether JNE could yield equivalent values for functionally distinct mappings due to scaling or dimensional effects. To address these issues systematically, this appendix presents a theoretical derivation of the conditions for JNE equivalence and an empirical evaluation on real fMRI data, demonstrating that such confounds have limited practical impact in neural encoding applications.

#### A.9.1  Theoretical Derivation

To analyze conditions under which two neurons yield equivalent JNE values, consider a simplified representation of the nonlinear encoding model. For the $i$-th sample, the predicted responses for two voxels are approximated as linear projections over transformed input features:

$$\hat{y}_{1,i} = \mathbf{h}_i^\top \mathbf{w}_1, \quad \hat{y}_{2,i} = \mathbf{h}_i^\top \mathbf{w}_2, \tag{13}$$

where $\mathbf{h}_i = [h(x_{1,i}), \ldots, h(x_{D,i})]^\top$ represents the nonlinear transformations of the $D$-dimensional input $\mathbf{x}_i$, and $\mathbf{w}_1, \mathbf{w}_2 \in \mathbb{R}^D$ are the learned output projection weights of the two neurons in the nonlinear encoding model.

The Jacobians (gradients) between inputs and outputs are then:

$$\mathbf{J}_{1,i} = \nabla_{\mathbf{x}_i} \hat{y}_{1,i} = \mathbf{g}_i^\top \mathbf{w}_1, \quad \mathbf{J}_{2,i} = \nabla_{\mathbf{x}_i} \hat{y}_{2,i} = \mathbf{g}_i^\top \mathbf{w}_2, \tag{14}$$

where $\mathbf{g}_i = [\partial h(x_{1,i})/\partial x_{1,i}, \ldots, \partial h(x_{D,i})/\partial x_{D,i}]^\top$ is the vector of local derivatives.

Assuming zero-mean gradients across $N$ samples (a simplification for analysis), JNE for each voxel is the standard deviation of the absolute Jacobian norms:

$$\text{JNE}_1 = \sqrt{\text{Var}\left(|\mathbf{J}_{1,1}|, \ldots, |\mathbf{J}_{1,N}|\right)}, \quad \text{JNE}_2 = \sqrt{\text{Var}\left(|\mathbf{J}_{2,1}|, \ldots, |\mathbf{J}_{2,N}|\right)}. \tag{15}$$

Let $\Sigma \in \mathbb{R}^{D \times D}$ be the covariance matrix of $\mathbf{g}$. Under the assumption that the entries of $\mathbf{g}$ follow a Gaussian distribution (approximating the central limit theorem in high dimensions), the absolute norm $|\mathbf{g}^\top \mathbf{w}|$ follows a folded normal distribution, yielding:

$$\text{JNE}_1 = \sqrt{1 - \frac{2}{\pi}} \cdot \sqrt{\mathbf{w}_1^\top \Sigma \mathbf{w}_1}, \quad \text{JNE}_2 = \sqrt{1 - \frac{2}{\pi}} \cdot \sqrt{\mathbf{w}_2^\top \Sigma \mathbf{w}_2}. \tag{16}$$

Thus, $\text{JNE}_1 = \text{JNE}_2$ holds if and only if $\mathbf{w}_1^\top \Sigma \mathbf{w}_1 = \mathbf{w}_2^\top \Sigma \mathbf{w}_2$. However, in real-world neural encoding models, the fully connected architecture typically induces complex dependencies across input dimensions. As a result, $\Sigma$ is generally a non-identity, non-diagonal matrix that includes substantial off-diagonal coupling terms. In such scenarios, even if $|w_1|_2 = \alpha|w_2|_2$, it does not guarantee a linear scaling relation $\text{JNE}_1 = \alpha \cdot \text{JNE}_2$, because the contribution of each input direction to JNE is modulated by the covariance structure.

More critically, functional heterogeneity across neurons leads to divergent encoding goals, input sensitivities, and nonlinear response characteristics. These structural constraints imply that the output weights $w_1$ and $w_2$—learned through training—are unlikely to align in direction or magnitude, making the assumption $w_1 = \alpha w_2$ almost never satisfied in practice. Thus, while JNE conflates response gain with nonlinearity in theory, such confounds have limited practical impact due to architectural and functional constraints.

### A.9.2 Empirical Evaluation on Real Data

To quantify the practical extent of these theoretical sensitivities, we implemented four variants of JNE and applied them to the NSD dataset [30] for S1, computing voxel-wise nonlinearity across significant visual cortex voxels (as defined in Section 3.3). These variants isolate the effects of norm choice and scale normalization, allowing us to assess stability under different formulations:

- **JNE-L1**: Original (centered Jacobians with L1 norm contraction along the feature dimension).
- **JNE-L2**: Centered Jacobians with L2 norm, to evaluate stability across different norm types.
- **JNE-zscore-L1**: Z-score normalization of Jacobian entries across input dimensions before L1 norm, which theoretically mitigates confounding from response gain.
- **JNE-zscore-L2**: Z-score normalization before L2 norm, combining scale invariance with geometric symmetry.

Table 4: Pairwise Pearson correlations between JNE variants on NSD fMRI data.

| Variant | JNE-L1 | JNE-L2 | JNE-zscore-L1 | JNE-zscore-L2 |
|---|---|---|---|---|
| JNE-L1 | - | 1.00 | 0.95 | 0.92 |
| JNE-L2 | - | - | 0.94 | 0.92 |
| JNE-zscore-L1 | - | - | - | 0.99 |
| JNE-zscore-L2 | - | - | - | - |

Pairwise Pearson correlation coefficients between the resulting JNE maps are shown in Table 4. All correlations exceed 0.9, indicating high consistency despite variations in norm and scaling. For example, the original JNE-L1 correlates at $r = 1.00$ with JNE-L2, suggesting that directional biases (e.g., L1 vs. L2 in rotationally invariant tunings) do not substantially alter regional patterns. Z-score variants, which mitigate pure gain effects, still align closely ($r \geq 0.90$) with unnormalized ones, confirming that scale sensitivity has minimal impact on relative comparisons in this dataset. These findings align with the theoretical analysis, underscoring JNE's robustness for interpretability in neural encoding applications.

### A.10 Theoretical Description of JNE-SS

JNE fundamentally characterizes linear or nonlinear properties by quantifying the standard deviation of $s\Delta\mathbf{JM}$ across different samples at each voxel, which reflects the variability of its mapping relationships across different samples. Further functional analysis of the standard deviation term $(s\Delta\mathbf{JM}_{i,m} - \mu_{\Delta,m})^2$ reveals that

$$\frac{1}{N} \sum_{i=1}^{N} (s\Delta\mathbf{JM}_{i,m} - \Delta\mu_m)^2 = \mathbb{E}_{\mathcal{S}} \left[ (s\Delta\mathbf{JM}_{\cdot,m} - \mathbb{E}_{\mathcal{S}}[s\Delta\mathbf{JM}_{\cdot,m}])^2 \right]$$

is essentially a second-order moment operator over the sample space $\mathcal{S} \subset \mathbb{R}^N$, quantifying the covariance structure of the Jacobian matrix within the sample distribution. Its non-zero value directly indicates deviations from the principle of linear superposition.

When the neural system is strictly linear, the Jacobian matrix $\mathbf{JM}_i$ of the mapping function $f : x \mapsto \hat{y}$ remains invariant across samples ($\mathbf{JM}_i \equiv \mathbf{JM}_{mean}$), causing the sample subspace $\{s\Delta\mathbf{JM}_i\}$ to collapse to the origin, leading to $\text{JNE}_m \equiv 0$. However, nonlinear mechanisms disrupt this invariance, as the local linear approximation ($\mathbf{JM}_i$) of a nonlinear system varies with the position of input $x_i$ on the manifold, introducing sample-dependent deviations in $\Delta s\mathbf{JM}_i$.

In this context, the squared deviation term $(s\Delta\mathbf{JM}_{i,m} - \Delta\mu_m)^2$ serves as a **microscopic measure of Neural Compliance**, capturing the variability of neural encoding at each sample level:

• **Low compliance (small deviation):** Voxel $m$ adheres to the linear superposition principle in its response to stimulus changes, exhibiting high stability in neural response.

• **High compliance (large deviation):** Voxel $m$ is governed by higher-order nonlinear mechanisms, with its information transmission exhibiting critical fluctuations.

Thus, $(s\Delta\mathbf{JM}_{i,m} - \Delta\mu_m)^2$ quantifies the sample-dependent dispersion of encoding at an individual voxel level, providing insights into the variability of neural responses across samples. Based on this, we define a novel metric for quantifying sample-specific nonlinear properties at each voxel, termed **JNE** with **S**ample-**S**pecificity (**JNE-SS**), formally expressed as

$$\mathbf{JNE}\text{-}\mathbf{SS} = (s\Delta\mathbf{JM} - \Delta\mu)^2 \in \mathbb{R}^{N \times M}$$

### A.11   Dimensionality Reduction, Ordering, and Classification Based on t-SNE

To systematically characterize the trend of voxel-wise nonlinear response properties across stimulus samples—that is, to identify which samples elicit more linear versus more nonlinear neural responses—we first applied t-SNE [53] to reduce the dimensionality of CLIP-ViT final-layer representations for all 73,000 stimulus images. Based on the resulting 1D t-SNE embedding, the image samples were then sorted in descending order, and the test set samples were selected accordingly. Since each subject's test set consisted of the same 1,000 images shared across the NSD dataset, this image ordering remained consistent across participants. Fig. 10 presents the sorted images, revealing a smooth transition and the clustering of semantically similar samples.

Subsequently, we performed K-means clustering on the 2D t-SNE representation (with the optimal number of clusters selected via silhouette analysis over the range 2–100; see Fig. 11). The 1,000 test images were grouped into 43 clusters, which were then organized according to the 1D t-SNE order. Notably, we observed that these 43 clusters also exhibited a smooth transitional structure (Fig. 12).

### A.12   Quantitative Comparisons across Regions

Table 5 provides the unnormalized MoM values used in the sample-specific analysis (Section 3.5), enabling quantitative comparisons across regions without the visual bias introduced by within-ROI normalization in Fig. 8c.

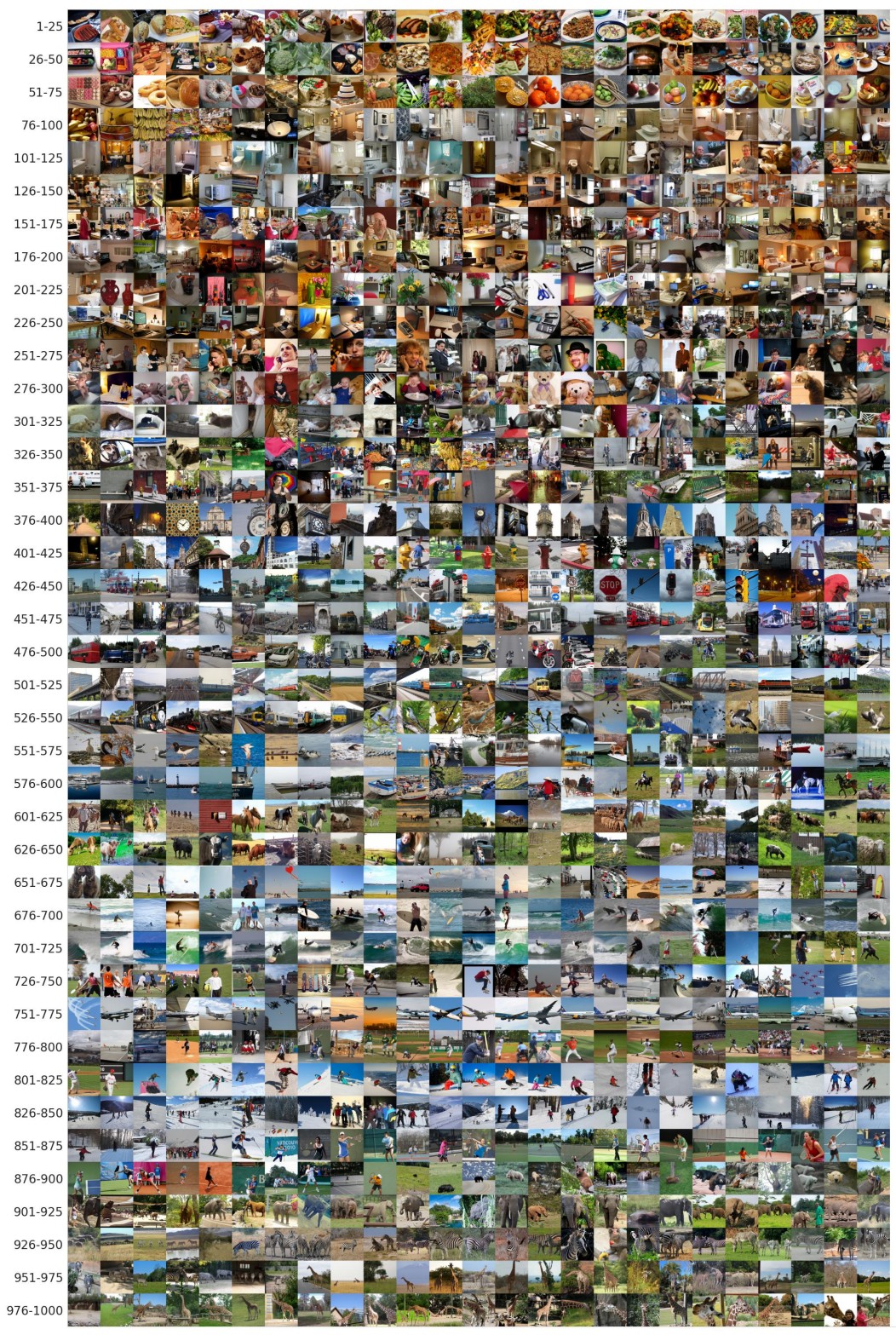

Figure 10: **Sample ranking based on t-SNE (1D dimensionality reduction, sorted in ascending order) and visualization of the ranked test set (consistent across subjects).**

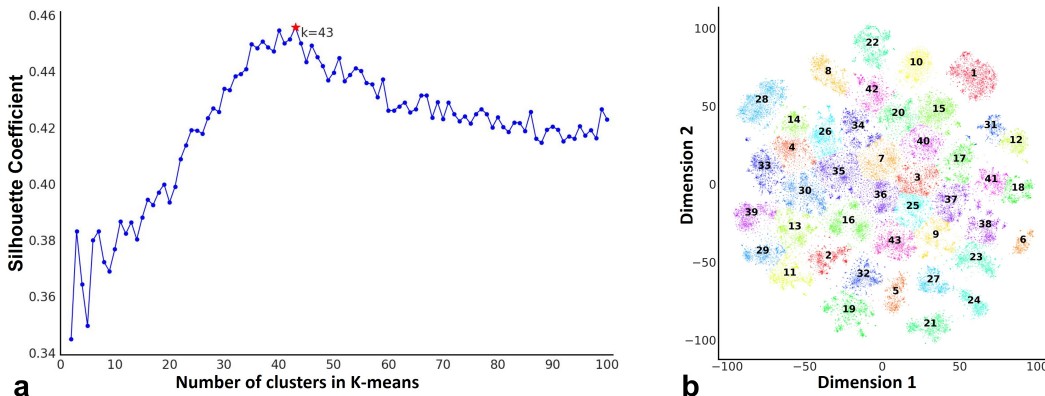

Figure 11: **Optimal cluster selection for K-Means based on the silhouette coefficient method (range: 2-100) and 2D dimensionality reduction visualization using t-SNE (with K-Means clustering and cluster number=43)**.

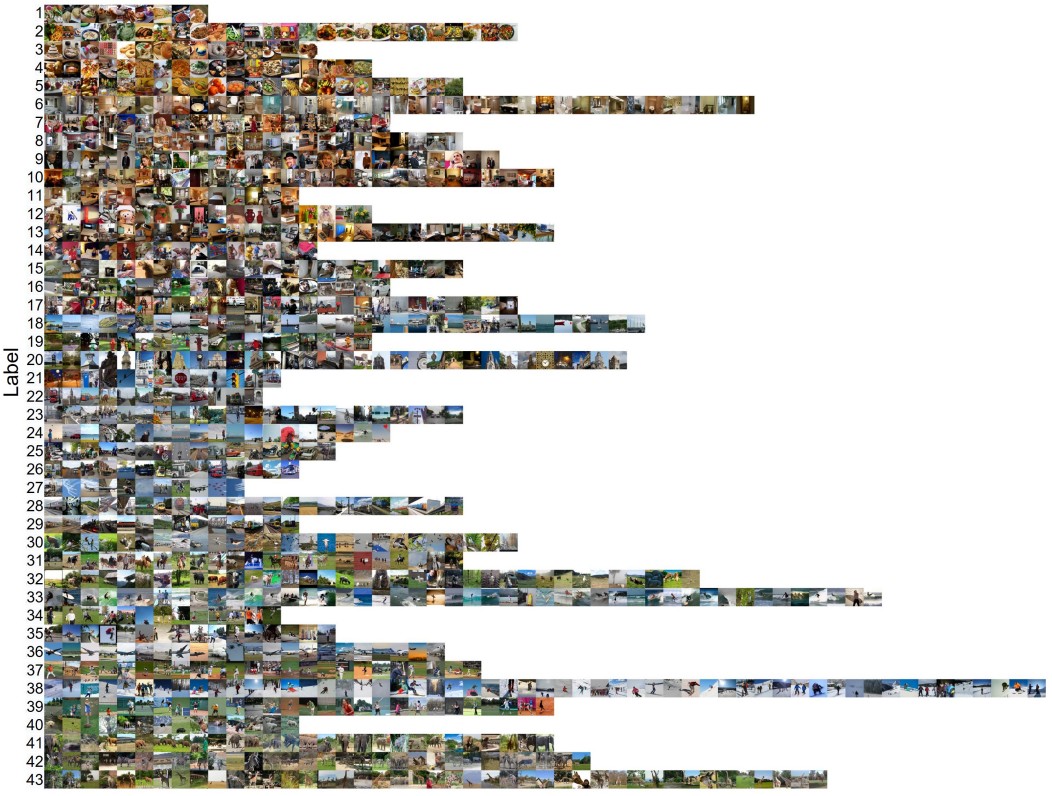

Figure 12: **Image category ranking based on K-Means clustering (number of clusters = 43)**.

Table 5: Raw Mean of MoJ (MoM) values across brain regions and image categories.

| Categories | Brain | V1 | V2 | V3 | V4 | EBA | PPA | RSC | OPA | FFA-1 | FFA-2 |
|---|---|---|---|---|---|---|---|---|---|---|---|
| C1 | 0.016 | 0.012 | 0.012 | 0.014 | 0.018 | 0.019 | 0.020 | 0.024 | 0.020 | 0.015 | 0.015 |
| C2 | 0.017 | 0.016 | 0.016 | 0.016 | 0.023 | 0.021 | 0.017 | 0.020 | 0.021 | 0.017 | 0.018 |
| C3 | 0.017 | 0.013 | 0.013 | 0.013 | 0.017 | 0.021 | 0.020 | 0.030 | 0.017 | 0.016 | 0.017 |
| C4 | 0.021 | 0.015 | 0.014 | 0.016 | 0.019 | 0.032 | 0.017 | 0.019 | 0.018 | 0.031 | 0.040 |
| C5 | 0.018 | 0.019 | 0.018 | 0.017 | 0.021 | 0.019 | 0.025 | 0.034 | 0.020 | 0.016 | 0.018 |
| C6 | 0.014 | 0.012 | 0.012 | 0.013 | 0.013 | 0.018 | 0.020 | 0.015 | 0.019 | 0.014 | 0.017 |
| C7 | 0.022 | 0.016 | 0.016 | 0.016 | 0.015 | 0.028 | 0.018 | 0.020 | 0.017 | 0.023 | 0.030 |
| C8 | 0.014 | 0.011 | 0.011 | 0.012 | 0.014 | 0.022 | 0.019 | 0.013 | 0.016 | 0.015 | 0.023 |
| C9 | 0.023 | 0.018 | 0.018 | 0.018 | 0.019 | 0.035 | 0.022 | 0.025 | 0.026 | 0.032 | 0.043 |
| C10 | 0.015 | 0.012 | 0.012 | 0.012 | 0.014 | 0.020 | 0.019 | 0.016 | 0.016 | 0.014 | 0.019 |
| C11 | 0.015 | 0.012 | 0.012 | 0.012 | 0.014 | 0.020 | 0.017 | 0.016 | 0.016 | 0.014 | 0.017 |
| C12 | 0.023 | 0.021 | 0.020 | 0.020 | 0.025 | 0.030 | 0.023 | 0.024 | 0.024 | 0.021 | 0.028 |
| C13 | 0.016 | 0.013 | 0.013 | 0.015 | 0.018 | 0.021 | 0.019 | 0.016 | 0.018 | 0.016 | 0.021 |
| C14 | 0.048 | 0.018 | 0.018 | 0.021 | 0.020 | 0.067 | 0.017 | 0.020 | 0.016 | 0.062 | 0.095 |
| C15 | 0.021 | 0.020 | 0.018 | 0.021 | 0.026 | 0.032 | 0.018 | 0.014 | 0.020 | 0.023 | 0.032 |
| C16 | 0.019 | 0.011 | 0.011 | 0.013 | 0.013 | 0.035 | 0.016 | 0.015 | 0.016 | 0.027 | 0.038 |
| C17 | 0.020 | 0.013 | 0.012 | 0.014 | 0.014 | 0.032 | 0.018 | 0.017 | 0.016 | 0.024 | 0.027 |
| C18 | 0.017 | 0.012 | 0.012 | 0.013 | 0.015 | 0.025 | 0.019 | 0.018 | 0.018 | 0.018 | 0.023 |
| C19 | 0.018 | 0.012 | 0.012 | 0.014 | 0.015 | 0.026 | 0.016 | 0.015 | 0.017 | 0.019 | 0.024 |
| C20 | 0.015 | 0.012 | 0.012 | 0.013 | 0.014 | 0.023 | 0.018 | 0.018 | 0.017 | 0.014 | 0.018 |
| C21 | 0.017 | 0.015 | 0.014 | 0.016 | 0.020 | 0.021 | 0.022 | 0.024 | 0.024 | 0.015 | 0.016 |
| C22 | 0.017 | 0.011 | 0.012 | 0.013 | 0.015 | 0.025 | 0.023 | 0.019 | 0.019 | 0.015 | 0.020 |
| C23 | 0.016 | 0.012 | 0.012 | 0.013 | 0.015 | 0.022 | 0.024 | 0.025 | 0.022 | 0.015 | 0.020 |
| C24 | 0.021 | 0.018 | 0.016 | 0.016 | 0.017 | 0.032 | 0.018 | 0.015 | 0.017 | 0.022 | 0.030 |
| C25 | 0.021 | 0.015 | 0.016 | 0.018 | 0.021 | 0.030 | 0.025 | 0.023 | 0.023 | 0.020 | 0.025 |
| C26 | 0.021 | 0.017 | 0.016 | 0.018 | 0.023 | 0.034 | 0.025 | 0.026 | 0.026 | 0.017 | 0.023 |
| C27 | 0.015 | 0.011 | 0.011 | 0.012 | 0.014 | 0.021 | 0.020 | 0.019 | 0.019 | 0.015 | 0.022 |
| C28 | 0.015 | 0.012 | 0.012 | 0.013 | 0.014 | 0.020 | 0.016 | 0.015 | 0.015 | 0.014 | 0.021 |
| C29 | 0.015 | 0.012 | 0.012 | 0.013 | 0.015 | 0.018 | 0.019 | 0.016 | 0.016 | 0.014 | 0.018 |
| C30 | 0.015 | 0.011 | 0.012 | 0.012 | 0.013 | 0.021 | 0.018 | 0.016 | 0.016 | 0.016 | 0.022 |
| C31 | 0.019 | 0.013 | 0.013 | 0.014 | 0.015 | 0.031 | 0.021 | 0.020 | 0.020 | 0.027 | 0.034 |
| C32 | 0.023 | 0.014 | 0.014 | 0.016 | 0.017 | 0.041 | 0.022 | 0.020 | 0.020 | 0.029 | 0.040 |
| C33 | 0.018 | 0.011 | 0.012 | 0.013 | 0.014 | 0.027 | 0.019 | 0.018 | 0.018 | 0.021 | 0.027 |
| C34 | 0.040 | 0.015 | 0.016 | 0.023 | 0.023 | 0.097 | 0.028 | 0.025 | 0.025 | 0.050 | 0.091 |
| C35 | 0.021 | 0.011 | 0.011 | 0.013 | 0.013 | 0.041 | 0.015 | 0.016 | 0.016 | 0.022 | 0.043 |
| C36 | 0.015 | 0.011 | 0.011 | 0.012 | 0.014 | 0.026 | 0.016 | 0.016 | 0.016 | 0.018 | 0.026 |
| C37 | 0.027 | 0.013 | 0.013 | 0.016 | 0.016 | 0.065 | 0.016 | 0.015 | 0.015 | 0.029 | 0.048 |
| C38 | 0.020 | 0.011 | 0.011 | 0.013 | 0.014 | 0.035 | 0.017 | 0.016 | 0.016 | 0.023 | 0.032 |
| C39 | 0.023 | 0.010 | 0.011 | 0.013 | 0.014 | 0.057 | 0.018 | 0.016 | 0.016 | 0.027 | 0.052 |
| C40 | 0.017 | 0.012 | 0.013 | 0.014 | 0.016 | 0.027 | 0.019 | 0.018 | 0.018 | 0.018 | 0.021 |
| C41 | 0.020 | 0.014 | 0.014 | 0.016 | 0.016 | 0.037 | 0.018 | 0.017 | 0.018 | 0.026 | 0.036 |
| C42 | 0.020 | 0.013 | 0.013 | 0.015 | 0.016 | 0.031 | 0.023 | 0.019 | 0.019 | 0.022 | 0.029 |
| C43 | 0.022 | 0.015 | 0.015 | 0.017 | 0.018 | 0.037 | 0.020 | 0.020 | 0.021 | 0.028 | 0.037 |

# B  Appendix B

## B.1  Supplementary Results on Comparison Between Linear and Nonlinear Neural Encoding Models

This section presents results for other subjects (S2, S5, S7). Consistent with the findings reported in the main text, we observed in Fig. 13 that both linear and nonlinear neural encoding models exhibited similar spatial patterns of predictive performance across the whole brain, with peak accuracy localized in visual cortices and several higher-order association areas. These results suggest that, regardless of inter-individual variability, both models reliably capture neural response patterns in the visual cortex.

To further quantify the spatial differences in predictive performance between the two models, we projected the $R^2$ scores of the linear and nonlinear models onto a common cortical surface and computed voxel-wise difference maps (Fig. 14). In these maps, white regions indicate comparable performance, red indicates higher accuracy for the nonlinear model, and blue indicates the opposite. The predominance of white regions suggests that both models achieve highly similar predictive accuracy at the voxel level across the brain, with only subtle differences localized in higher-order cortical areas, potentially reflecting nonlinear processing of abstract or semantic information in those regions.

We also performed Pearson correlation analyses on the $R^2$ sequences from the linear and nonlinear models, both across the whole brain and within ten visual-related ROIs. As shown in Fig. 15, all correlations were significantly positive across subjects and ROIs, with the lowest correlation being $r = 0.86$ in V1 of S5 ($P < 0.001$). These results further confirm the strong consistency in predictive

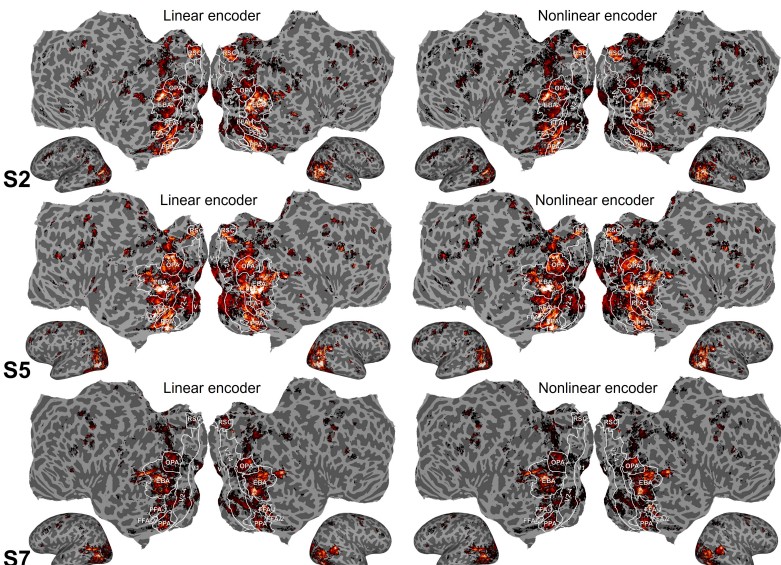

Figure 13: **Supplementary results comparing the predictive performance of linear and nonlinear neural encoding models.** This figure presents the prediction performance of both linear and nonlinear neural encoding models for S2, S5, and S7. The colorbar ranges in all supplementary figures are consistent with those used in the main text.

capacity between the two models and indicate that, when driven by deep visual features extracted from CLIP-ViT, both models characterize fMRI responses in a highly similar manner.

From a neuro-computational perspective, these findings suggest that linear neural encoding models are already sufficient to capture the neural response structure in the visual cortex when using CLIP-ViT features. While nonlinear models offer greater representational flexibility, they do not yield systematic gains in predictive performance under these conditions. Therefore, relying solely on differences in $R^2$ scores between linear and nonlinear models is not a reliable strategy for inferring the presence of nonlinear neural response characteristics—particularly when using features that already integrate high-level semantic information.

In conclusion, the cross-subject analysis supports our main-text finding: the difference in $R^2$ between linear and nonlinear models cannot robustly reflect voxel-level nonlinear properties. This provides a theoretical motivation for the development of our JNE metric, which is designed to reveal fine-grained structure in neural responses beyond what conventional prediction-based comparisons can offer.

### B.2 Supplementary Results on Cortical Distribution of Nonlinearity and Hierarchical Effects in Visual Cortex

Fig. 16 presents the cortical distribution of BOLD response nonlinearity in additional subjects (S2, S5, S7), further validating the generalizability and consistency of this property across individuals. Consistent with S1, high JNE values were predominantly localized in higher-order association cortices, particularly at the temporoparietal-occipital junction (TPOJ), ventral temporal lobe, and medial prefrontal cortex. These regions have long been implicated in mul-timodal semantic integration, higher-order cognition, and social reasoning, and are known for their high selectivity to abstract features such as semantics, identity, and context [2, 30, 35, 39, 40, 41]. The observed higher variability in neural responses across different stimulus conditions in these regions likely reflects stronger nonlinear characteristics.

Fig. 17 further presents the average JNE values across different hierarchical levels of the visual cortex. Across all four subjects, higher-level visual areas consistently exhibited the highest JNE values, while primary visual areas showed significantly lower values, with intermediate areas falling in between. A bar plot comparison in Fig. 18 illustrates this pattern, highlighting that higher-level visual cortices not only have higher mean JNE values but also a greater proportion of voxels with high JNE. This pattern

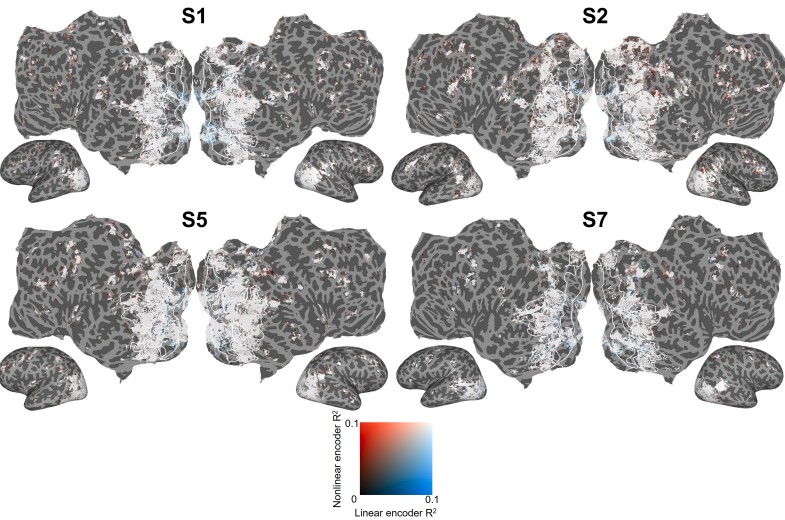

Figure 14: **Mapping of prediction performance for linear and nonlinear neural encoding models on the same cortical atlas.** White regions indicate voxels where both models perform similarly; blue regions indicate voxels where the linear neural encoding model outperforms the nonlinear model; red regions indicate the opposite.

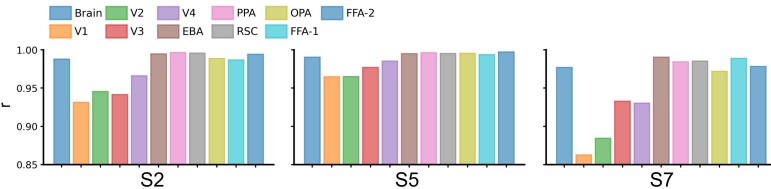

Figure 15: **Bar plots of Pearson correlation coefficients of $R^2$ values across the whole brain and 10 ROIs for S2, S5, and S7.**

supports the presence of a hierarchical gradient in BOLD response nonlinearity within the visual system: as the level of information processing increases, neural response variability and nonlinearity also increase.

Fig. 19 further analyzes the distribution of JNE values, presenting the density curves across primary, intermediate, and higher-level visual areas. We observed that JNE values in primary areas are concentrated at the lower end, while intermediate areas show a broader and slightly right-skewed distribution, and higher-level areas exhibit a clear right-skew with a long-tailed structure. Collectively, Figs. 17, 18, 19, and 20 consistently demonstrate a hierarchical increase in nonlinearity across the visual cortical hierarchy.

Notably, in Fig. 20, the intermediate visual cortex of S2 did not show markedly higher JNE levels compared to the PVC. To clarify this, we further examined JNE values in V1, V2, V3, and V4 of S5, and found that V4 exhibited significantly higher JNE values than V1–V3 ($P < 0.001$). This finding still supports our general conclusion that BOLD response nonlinearity increases with cortical hierarchy and suggests that even in the presence of individual variability, hierarchical patterns of nonlinearity remain evident within specific visual subregions.

In summary, cross-subject analyses reinforce the conclusion that BOLD response nonlinearity is not uniformly distributed across the cortex. Instead, it shows pronounced spatial clustering and functional specificity, particularly in higher-order association areas involved in mul-timodal integration and semantic abstraction. Furthermore, nonlinearity within the visual pathway follows a clear hierarchical organization, with systematic increases from primary to higher-level cortices.

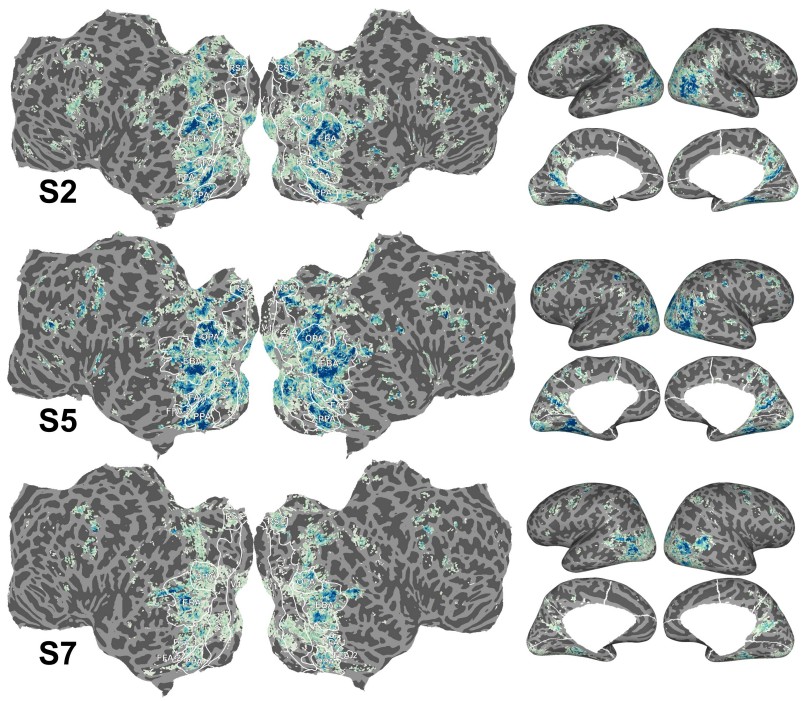

Figure 16: **Spatial distribution of BOLD response nonlinearity in S2, S5, and S7.**

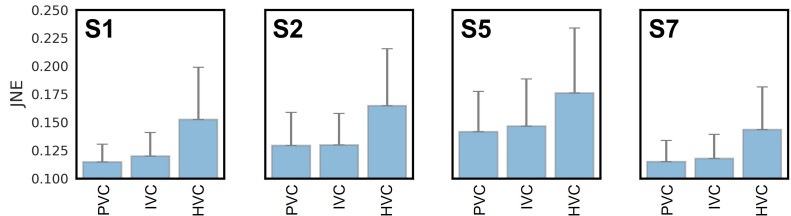

Figure 17: **Comparison of JNE statistical metrics (mean, standard deviation) across primary visual cortex (PVC), intermediate visual cortex (IVC), and higher-order visual cortex (HVC) for S1, S2, S5, and S7.** Primary visual cortex (PVC) - V1; Intermediate visual cortex (IVC) - V2, V3, V4; Higher-order visual cortex (HVC) - EBA, PPA, RSC, OPA, FFA-1, FFA-2.

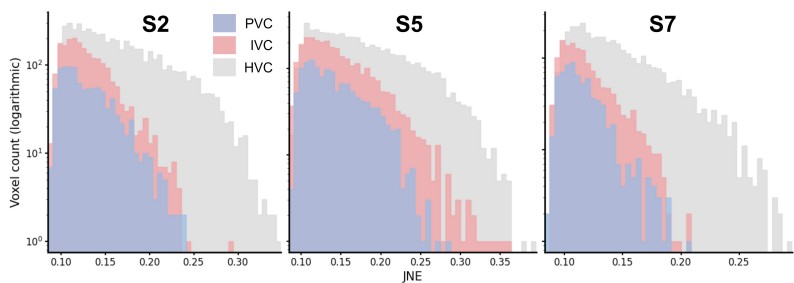

Figure 18: **The bar plot comparison of JNE in primary, intermediate, and higher-order visual cortex for S2, S5, and S7.**

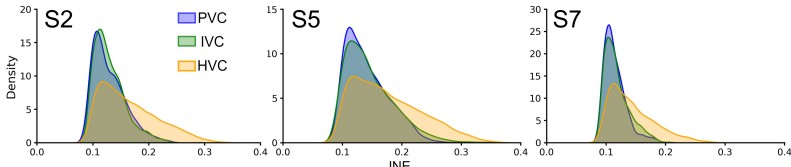

Figure 19: **The comparison of JNE distribution maps across primary visual cortex, intermediate visual cortex, and higher-order visual cortex for S2, S5, and S7.**

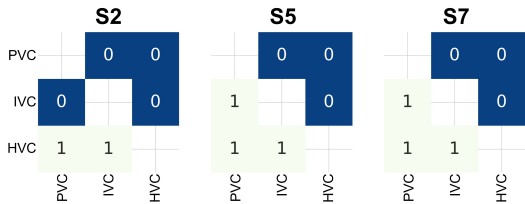

Figure 20: **Significance heatmap of the primary visual cortex, intermediate visual cortex, and higher-order visual cortex for pairs of regions (S2, S5, S7).** In the heatmap, a value of 1 at position (i, j) indicates that the JNE of brain region i is significantly greater than that of brain region j, while a value of 0 indicates no significant difference.

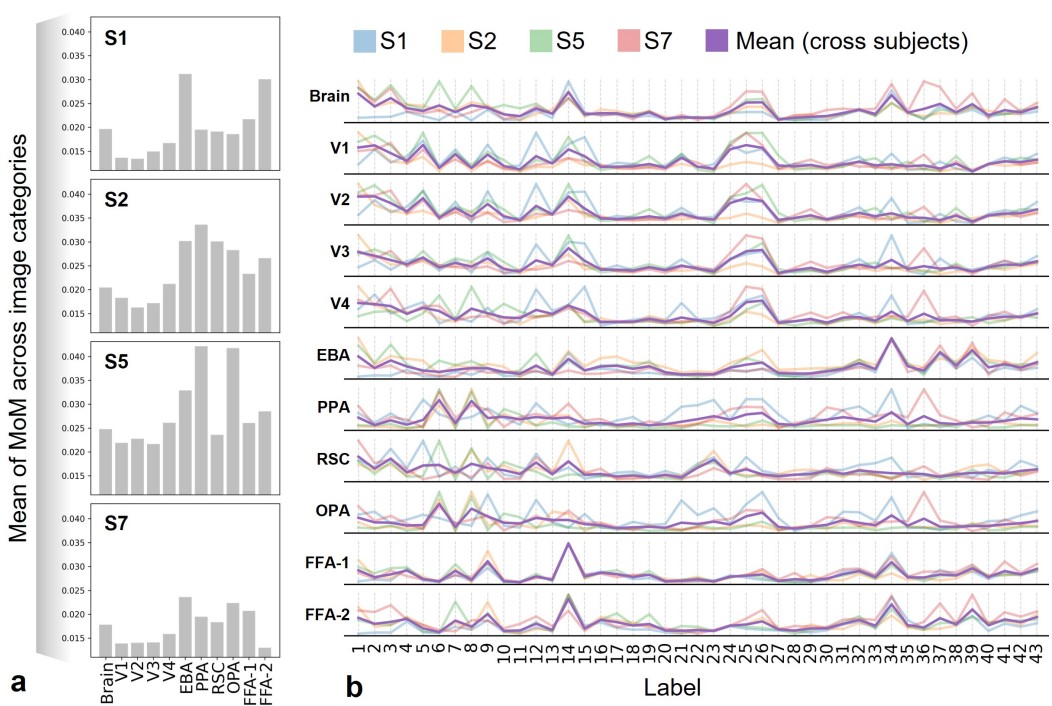

Figure 21: **Statistical (mean) display of MoM values across the whole brain and 10 ROIs, along with the nonlinear category preferences of S1, S2, S5, and S7. a**. Shows the MoM values across the whole brain and 10 ROIs for the four subjects, reflecting the spatial distribution of MoM across different brain regions; **b**. Displays the max-min normalized MoM (NMoM) statistical results for the nonlinear response category preferences across the four subjects, with the average values of the four subjects shown to characterize statistically significant selective patterns for different categories in nonlinear encoding.

### B.3 Supplementary Results on Sample-Specific Preferences of Nonlinear Neural Responses

We further analyzed the sample-specific preferences of nonlinear neural response characteristics across different subjects, as shown in Fig. 21. Specifically, Fig. 21a illustrates the mean MoM values across all image categories within each visual cortical region for the four subjects. This result further supports the existence of significant spatial inhomogeneity in nonlinear response characteristics across the cortex. An overall trend emerged, suggesting that nonlinearity increases progressively along the visual hierarchy, i.e., PVC < IVC < HVC. This hierarchical enhancement of nonlinearity aligns well with classical models of visual processing [36, 37, 38], indicating that neural responses tend to become increasingly nonlinear as the complexity of information processing increases.

Although the overall pattern is largely consistent, we also observed individual variability across subjects. For instance, in S7, the mean JNE value of FFA-2, a higher-order visual area, was unexpectedly lower than that of the primary and intermediate visual cortices (Fig.21a). This may reflect subject-specific structural or functional differences in facial stimulus processing. Interestingly, in the same subject, we observed a notably high JNE level in FFA-1, suggesting that within the same functional module, different subregions may show complementary or reconfigurable selectivity to specific stimulus categories. Such a pattern may arise from the heterogeneous microcircuitry within cortical areas and the diverse semantic feature dimensions being encoded.

Fig. 21b further reveals the category-level selectivity of nonlinear responses. We found that higher-order visual regions exhibited peak nonlinear responses for certain image categories, such as portraits, buildings, and vehicles—stimuli that often contain rich structural and semantic content and thus require more complex perceptual and integrative processes. In contrast, the primary visual cortex showed relatively high nonlinear responses across a broader range of image categories, suggesting that although these early regions are generally considered more linear in processing low-level visual features (e.g., edges, textures, local contrast), the richness of natural stimuli can still elicit complex and possibly nonlinear response patterns due to contextual combinations or stimulus-specific configurations. Overall, these findings complement the results in Fig. 8.

