# OpenReview forum: "Jacobian-Based Interpretation of Nonlinear Neural Encoding Model"
_NeurIPS.cc/2025/Conference — NeurIPS 2025 spotlight_

### Official Review · Reviewer_cJqm · 2025-06-23

**Clarity:** 4
**Significance:** 3
**Originality:** 3
**Rating:** 5
**Confidence:** 4

**Summary:**

This paper introduces the Jacobian-based nonlinearity evaluation (JNE), a metric for quantifying the nonlinearity of neural encoding models.  The authors first argue that the common practice of comparing the predictive accuracy ($R^2$) of linear versus nonlinear models is an unreliable method for assessing nonlinearity, especially when using pre-trained features.  The proposed JNE metric quantifies the statistical variability of a model's local linear mappings (its Jacobian matrices) across a diverse set of input stimuli.  Applying this metric to fMRI data reveals two primary contributions: first, it provides quantitative evidence for a hierarchical gradient of increasing nonlinearity from the primary visual cortex to higher-order visual areas.  Second, it demonstrates that this nonlinearity is stimulus-selective, with regions such as the fusiform face area (FFA) and the parahippocampal place area (PPA) showing the most pronounced nonlinear responses to their preferred stimulus categories (faces and scenes, respectively), thereby linking a region's computational properties to its known functional specialization.

**Questions:**

I suggest that you generate a large number of simulated linear and nonlinear networks, calculate the JNE score for each network and
treat "linear" and "nonlinear" as the two ground-truth classes and the JNE score as the classifier's output. Could you then plot an ROC curve to see how well the JNE score distinguishes between the two classes across all possible thresholds? The AUC would quantify the metric's discriminative power in this idealized environment.

**Ethical Concerns:**

["NO or VERY MINOR ethics concerns only"]

**Final Justification:**

I'm happy with the author's response and retain my score

**Limitations:**

The discussion of limitations, including the critical BOLD confound, should not be in the appendix, but this can be fixed in the camera-ready version.

**Quality:**

4

**Strengths And Weaknesses:**

The work is well-executed and addresses a timely and important problem

A major strength is the empirical demonstration that comparing the predictive accuracy ($R^2$) of linear and nonlinear models is an unreliable method for assessing nonlinearity. By showing that both model types achieve nearly identical performance when using powerful features from foundation models like CLIP-ViT, the authors effectively invalidate a common heuristic and establish a clear need for a new approach.

The JNE metric is conceptually novel. While the Jacobian matrix is a standard mathematical tool, the proposal to use the statistical dispersion of Jacobians across a sample population as a direct, non-parametric measure of nonlinearity is an insightful formulation.  Its simplicity is noteworthy when compared to more complex learnable frameworks like LinBridge, which require training an auxiliary model to decompose the Jacobian.

The JNE metric is built on the principle that a truly linear system must have an input-invariant Jacobian. The authors then rigorously validate this premise through a series of well-designed simulation experiments. The network-level validation further confirms that JNE can reliably detect and quantify nonlinearity in more complex, multi-layer systems.

The introduction clearly articulates the problem and motivates the need for the JNE metric. The four-step computational workflow of JNE is presented with mathematical precision and is well-supported by a clear diagram (Fig. 2a,c).

The most critical limitation is that the JNE metric is applied to a model of the fMRI BOLD signal, not directly to neural activity. The BOLD signal is an indirect and sluggish hemodynamic measure that has its complex nonlinear relationship with the underlying neural events. These nonlinearities can be vascular in origin (e.g., blood flow refractory effects) or related to neural phenomena like adaptation, and are highly sensitive to experimental design, such as stimulus timing.

The paper notes that JNE lacks a defined threshold to distinguish a "linear" from a "nonlinear" region in absolute terms. This restricts the metric's utility to relative comparisons. It could, however, be quantified with simulation experiments and an ROC analysis, at least in an idealized setting.

While the method is novel, the neuroscientific findings could be viewed as incremental. The discussion of limitations, including the critical BOLD confound, should not be in the appendix, but this can be fixed in the camera-ready version.

---

> ### Author Rebuttal · Authors · 2025-07-31
>
> We would like to thank the reviewer for the constructive comments and suggestions, as well as the appreciation of our study. Our point-by-point responses are detailed as follows.
>
> ### **Response to Weaknesses**
>
> **W1:** *The most critical limitation is that the JNE metric is applied to a model of the fMRI BOLD signal, not directly to neural activity. The BOLD signal is an indirect and sluggish hemodynamic measure that has its complex nonlinear relationship with the underlying neural events. These nonlinearities can be vascular in origin (e.g., blood flow refractory effects) or related to neural phenomena like adaptation, and are highly sensitive to experimental design, such as stimulus timing.*
>
> **Response:** We appreciate the reviewer's comments. As noted, the BOLD signal is an indirect and temporally filtered measurement of neural activity, shaped by complex and nonlinear hemodynamic processes.
>
> Accordingly, the scope of our analysis is explicitly limited to the nonlinear structure of BOLD signals as observed via fMRI. The goal of JNE is to quantify the overall degree of nonlinearity in the mapping from input stimuli to BOLD responses, within the model–observation space. We do not claim that JNE directly reflects neuronal-level nonlinearity. Rather, it should be viewed as a tool for quantifying the explanatory behavior of the encoding model.
>
> We also acknowledge that JNE does not distinguish whether this nonlinearity arises from neuronal or vascular mechanisms. Thus, JNE should be understood as an empirical estimate of the input–output nonlinearity of the composite function, whose interpretability is inherently constrained by the limits of fMRI as a measurement modality. In future work, we plan to incorporate HRF deconvolution strategies to attempt disentangling neural and vascular sources of nonlinearity. Furthermore, we will validate the generalizability of JNE to neural activity itself using data with higher temporal resolution (e.g., ECoG or MEG).
>
> We will carefully revise the relevant sections to ensure that "BOLD nonlinearity" and "neural nonlinearity" are clearly distinguished, and that all terminology is used with precision.
>
> **W2:** *The paper notes that JNE lacks a defined threshold to distinguish a "linear" from a "nonlinear" region in absolute terms. This restricts the metric's utility to relative comparisons. It could, however, be quantified with simulation experiments and an ROC analysis, at least in an idealized setting.*
>
> **Response:** We sincerely thank the reviewer for the valuable suggestion. In its current form, JNE is primarily used for relative comparison of nonlinearity levels across model architectures or brain regions in empirical analyses, and that a clear absolute decision boundary between "linear" and "nonlinear" systems has not yet been established.
>
> Follow the insightful suggestion in **Q4**, we have generated 382,340 simulated linear and nonlinear networks and performed a binary classification based on the JNE metric. The classifier achieved an ROC of 1, with linear neural networks exhibited 0 JNE values while nonlinear networks  were with non-zero JNE values. However, simulations under more complex conditions such as non-uniform distribution and data noise have not been accounted for due to time limit.
>
> Specifically, we have extended our original simulation framework (Section 3.1: *Network-level validation via ANN-based simulation*) to systematically explore the structural space of artificial networks, introducing "layer number" as a controlled variable $$, and jointly examining the following factors:
>
> * **Activation function type**: None, ReLU, LeakyReLU, GELU, Swish;
> * **Hidden layer width**: 16, 32, 64, 128, 256, 512, 1024;
> * **Residual connection configuration**: on / off;
> * **Activation function positions**: all $2^L$ combinations of insertion patterns per layer, where $L$ is the number of layers.
>
> By exhaustively combining these configurations, we generated 382,340 structurally diverse networks. For each network, we performed forward passes using a fixed input distribution $\mathcal{U}(-10, 10)$, computed the Jacobian between input and output, and extracted the JNE value.
>
> To evaluate the discriminative power of JNE, we formulated a binary classification task:
>
> * Networks with **no nonlinearities** (activation type = None, or all insert positions set to 0) were labeled as **linear** (label = 0);
> * All other networks were labeled as **nonlinear** (label = 1).
>
> **Simulation Results:** Under these idealized conditions, JNE demonstrated excellent discriminative performance, with an AUC of 1.0.
>
> **W3:** *While the method is novel, the neuroscientific findings could be viewed as incremental. The discussion of limitations, including the critical BOLD confound, should not be in the appendix, but this can be fixed in the camera-ready version.*
>
> **Response:** We thank the reviewer's appreciation of the novely of our study. The core contribution of this study is introducing a novel metric—JNE—for quantifying the explanatory power of nonlinear neural encoding models. The incremental neuroscientific findings serve as part of the validation of the proposed method. Accordingly, we will explicitly strengthen the presentation of JNE as a methodological innovation, and move the discussion of limitations to the main text.
>
> ### **Response to Questions**
>
> **Q1:** *I suggest that you generate a large number of simulated linear and nonlinear networks, calculate the JNE score for each network and treat "linear" and "nonlinear" as the two ground-truth classes and the JNE score as the classifier's output. Could you then plot an ROC curve to see how well the JNE score distinguishes between the two classes across all possible thresholds? The AUC would quantify the metric's discriminative power in this idealized environment.*
>
> **Response:** Following the reviewer's suggestion, we expanded our simulation experiments by generating over 380,000 simulated linear and nonlinear neural networks with diverse architectures. We then constructed a binary classification task based on the JNE metric. We found that JNE exhibited strong discriminative power, achieving an AUC of 1.0 under this controlled setting. Simulated linear networks exhibit 0 JNE value, and nonlinear networks are with non-zero JNE values. However, simulations under more complex conditions such as non-uniform distribution and data noise have not been accounted for due to time limit. Please refer to our response to **W2** for details.
>
> ### **Response to Limitations**
>
> **L1:** *The discussion of limitations, including the critical BOLD confound, should not be in the appendix, but this can be fixed in the camera-ready version.*
>
> **Response:** We will move the discussion of limitations to the main text.

---

> > ### Comment · Reviewer_cJqm · 2025-08-07
> >
> > I thank the authors for their additional work and have no further questions. I will retain my score for acceptance.

---

> ### Author Response · Authors · 2025-08-08
> **Response to Reviewer cJqm**
>
> We sincerely thank you for your positive feedback and for recommending our work for acceptance. We greatly appreciate your constructive comments throughout the review process, which have helped us further improve the clarity and rigor of our manuscript.

---

### Official Review · Reviewer_VQtJ · 2025-06-26

**Clarity:** 4
**Significance:** 2
**Originality:** 3
**Rating:** 4
**Confidence:** 5

**Summary:**

The authors propose a non-linearity metric (Jacobian Non-linearity Evaluation – JNE) to characterize the non-linear response properties of the brain. The JNE for mapping Clip-VIT output activations to brain responses (from the Natural Scenes Dataset) showed increasing JNE scores from primary visual cortex, to intermediate (V2-V4), to higher level visual cortex (e.g., EBA, FFA, PPA, etc.). Moreover, JNE scores were higher for preferred categories in higher visual cortex (e.g., images of outdoor scenes evoked stronger JNE in PPA and RSC; images of people and faces evoked stronger JNE in FFA and EBA). The authors conclude that JNE characterizes the degree of non-linearity in nonlinear neural encoding models, and describes the degree of nonlinearity across different brain regions.

**Questions:**

1)	The JNE score measures the degree of non-linearity in the “mapping” from model-space to voxel-space. Is it the case that high JNE scores can be observed if-an-only-if the target space exhibits high non-linearity? If this is not a provable constraint on JNE scores, we should be cautious about interpreting the mapping-JNE scores as a measure of the encoding-target-space-nonlinearity.
2)	On this same topic, I’m not sure what we can/should conclude about the brain regions from these JNE scores. To illustrate my concern, let’s pretend that some Clip-ViT output units respond more to “profile-faces” and others to “frontal views of faces”, and we are predicting FFA responses (face-selective cortex). Given that feature-response preference, I assume the JNE score would be higher if you presented a diverse image set (mixture of profile vs. frontal views), and lower if you presented a homogenous set (frontal views only). The brain region itself has “the same non-linearity” no matter what image set you present, but the JNE scores would vary.
3)	One more question on interpretation. The fact that the linear and non-linear encoders show equivalent performance mapping between CLIP-output and brain responses, it really seems like JNE scores are capturing the non-linearity of the mapping (which itself seems like it would depend on the variability of the dataset along dimensions captured by the encoder; see previous question), rather than non-linearity of the brain-response per se.
4)	I found myself wondering whether/how the JNE scores would vary as a function of model-layer. Typically earlier model layers show a stronger alignment with primary and intermediate visual cortex, whereas later model layers show stronger alignment with higher visual cortex. To what extent do JNE scores depend on the model-layer that feeds into the encoding model? Is the low JNE score for primary visual cortex due to the mismatch between model-space (CLIP-output activations)?
5) In theory, with a powerful enough encoder, it should be possible to map even an early model layer to higher-visual cortex, presumably requiring greater non-linearity in the mapping. Does this mean higher JNE scores? If so, given that the target brain regions aren’t changing, that variability would imply that JNE is a measure of the non-linearity in the mapping, not the target brain region.
6)	It would be helpful if the authors could situate this work more fully in the context of what is known about the biological visual processing hierarchy, it’s known increase in complexity (particular selectivity profiles + invariances), and non-linearity in visual processing (providing some clarity on how this is measured and quantified in biological systems). It would also be helpful to relate the current approach/findings to what has been demonstrated in prior work, particularly work showing a hierarchical correspondence between model activations and brain responses (early model layers predicting early brain regions better, later model layers predicting later model layers better), which has also been proposed to reflect comparable non-linear stages in artificial and biological visual systems. What new insights do we gain from the current approach?
7)	If the goal is to characterize properties of the brain, independent of the encoding model, it seems important to present results for encoding models mapping multiple model layers (from early to late to output layers) and multiple models (various architectures, visual diets, and objectives) to ensure that the conclusions about the brain are general/consistent across layers and models. My guess is that there will be significant variability based on layer (and possibly model depending on the suite drawn from), and that more nuanced conclusions regarding non-linearities in the mapping from model-space to brain-space are required.
8)	In the Discussion the authors propose that the present results support a bottom-up processing mechanism in the brain’s visual system. I’m not sure I understand exactly what the authors mean here, and what the alternative is. The biological visual system is known to have ubiquitous feed-forward, feed-back, and skip connections, and fMRI responses are too sluggish to isolate purely feed-forward processing, so the signal being modeled here is necessarily a result of both feed-forward and feed-back processing. If the authors mean that the input signal (image) is the primary driver of responses along the ventral processing stream, I would agree with that, but I’m not sure the present results speak to that much.

I want to emphasize that I think this is very solid work and that the results are intriguing enough for me to think the authors are onto something here. The hierarchical progression of JNE scores with brain-depth, and the category-dependent JNE scores (faces in FFA; scenes in PPA) are very interesting results which suggest the measure has potential to provide some new insight. I’m just not sure we can safely interpret the JNE values as a measure of non-linear processing in the target brain region. If not, if they are a measure of the non-linearity in mapping model-layer-A to brain-region-B, then I’m not sure exactly what they tell us about biological visual processing. In that case, I think we need more results to learn something new, some interesting pattern across model layers or across model architectures or models trained on different objectives. And then we’d need a framework for thinking about what “non-linearity in the mapping between spaces” means. Does higher JNE mean two spaces are “less natively aligned” and thus less similar? Does the perfect “model-layer for brain region B” show low JNE or high JNE when mapped to that region? (maybe that could be answered by aligning human-to-human?).

**Ethical Concerns:**

["NO or VERY MINOR ethics concerns only"]

**Final Justification:**

The quality and rigor in the methods seems clear to me. I thought that caution/clarity on the interpretation was needed, and deeper integration with related neuroscience research would increase impact of the work. The authors seem to have a solid plan for both.

**Limitations:**

yes

**Quality:**

3

**Strengths And Weaknesses:**

Strengths:
- Well-written, clear presentation.
- Well-motivated, strong methodological approach.
- Intriguing results.

Weaknesses:
-  The authors aim to understand non-linearities in biological neural processing, but the methods measure non-linearities in the mapping between spaces (model-output-space and voxel-space), and it’s not clear the measure can speak directly to non-linearities in brain responses.
- The work is not fully situated in the visual neuroscience literature, and what we already know about non-linear hierarchical processing in visual cortex, and hierarchical correspondence between models and brains. Authors need to clarify what have we learned about biological vision that we didn’t already know from prior work…It can’t be that non-linearities increase as you progress in the visual processing stream (already established in prior neurophysiology work; already inferred in hierarchical correspondence between models and brains). The authors can argue that JNE provides the first direct measurement of this putative model/brain correspondence, but then it needs to be more clearly demonstrated that JNE scores are directly attributable to properties of the target-space (brain responses). If the JNE scores are model-dependent, or model-layer-dependent, then the JNE scores can only be understood as a measure of the non-linearity in the mapping from model-layer-A to brain-region-B (a property of the mapping between two spaces).

---

> ### Author Rebuttal · Authors · 2025-07-31
>
> We would like to thank the reviewer for the constructive comments and suggestions, as well as the appreciation of our study. Our point-by-point responses are detailed as follows.
>
> ### **Response to Weaknesses**
>
> **W1:** *It's not clear the measure can speak directly to non-linearities in brain responses.*
>
> **Response:** We agree that JNE does not directly reflect neural-level nonlinearity, as fMRI BOLD signals are shaped by both neural activity and vascular dynamics. Therefore, JNE reflects a composite nonlinearity from input to BOLD response. The estimated nonlinearity captured by JNE may arise from either neural encoding or BOLD-related factors (e.g., HRF nonlinearity, vascular saturation), which cannot be fully disentangled. While BOLD signals are known to introduce complex nonlinearities, techniques such as HRF deconvolution or explicit modeling may help mitigate their influence. We thus position JNE as a practical tool for probing nonlinear encoding in fMRI settings, and will clarify its methodological scope and limitations more explicitly in the revised manuscript.
>
> **W2:** *The work is not fully situated in the visual neuroscience literature...*
>
> **Response:** We will revise the manuscript to emphasize methodological contributions and better situate our work within the visual neuroscience literature, especially regarding cortical hierarchies.
>
> ### **Response to Questions**
>
> **Q1.** *The JNE score measures...*
>
> **Response:** High JNE scores arise only when the target mapping exhibits substantial nonlinearity, as supported by our theoretical and simulation-based analyses. However, due to neural-vascular coupling, JNE cannot isolate cortical nonlinearity from BOLD signal properties. We therefore position JNE as a methodological tool for analyzing nonlinear encoding models, with fMRI serving as partial validation. In the revision, we will clarify this conceptual scope and explicitly discuss the limitations of interpreting JNE as a direct measure of neural nonlinearity.
>
> **Theoretical Justification**
>
> The JNE metric quantifies a model's sensitivity to input perturbations, defined as the standard deviation of the Jacobian norm computed across different input samples (i.e., predicted fMRI responses). By the chain rule, the complete input–output mapping in a neural encoding framework can be expressed as:
>
> $$
> I \xrightarrow{m(\cdot)} X \xrightarrow{f(\cdot)} \hat{y}
> $$
>
> Here, $m(\cdot)$ denotes a pretrained ANN that extracts features $X$ from the input $I$, and $f(\cdot)$ is a linear or nonlinear encoder that maps the features into the BOLD response space.
>
> Applying the chain rule gives:
>
> $$
> \frac{\partial \hat{y}}{\partial I} = \frac{\partial f(X)}{\partial X} \cdot \frac{\partial X}{\partial I}
> $$
>
> Since the feature extractor $m(\cdot)$ is fixed, the term $\partial X / \partial I$ is constant across all voxels. Therefore, voxel-wise differences in input–output sensitivity are entirely determined by $\partial f(X) / \partial X$. From a physiological perspective, the actual brain response pathway can be approximated as:
>
> $$
> I \xrightarrow{g(\cdot)} \text{Neural Activity} \xrightarrow{z(\cdot)} y
> $$
>
> If the ANN representation $X = m(I)$ partially approximates the neural state (a widely supported assumption in prior neural encoding work), then $\partial f(X) / \partial X$ can be viewed as a first-order approximation of $\partial z(g(I)) / \partial I$. That is:
>
> $$
> \frac{\partial f(X)}{\partial X} \propto \frac{\partial z(g(I))}{\partial I}
> $$
>
> Thus, variation in the JNE score reflects the degree of nonlinear sensitivity in the composite system $z(g(\cdot))$. JNE only increases when this system exhibits substantial input-dependent nonlinearity.
>
> **Simulations**
>
> To validate the above derivation, we designed a structured simulation experiment:
>
> * We constructed a tensor $A \in \mathbb{R}^{1000 \times 244 \times 768}$ to simulate the ANN-based input-to-feature mapping;
> * We defined a tensor $B \in \mathbb{R}^{1000 \times 768 \times 1}$ to represent nonlinear mappings from features to response;
> * We sampled 100 scaling coefficients $\lambda_k = \sin(x_k)$ uniformly from $[0, 2\pi]$, simulating amplitude-modulated responses;
> * The final output was computed as $C_k = \texttt{einsum}(A, \lambda_k B)$, simulating input-dependent model responses.
>
> We then computed JNE for both $\lambda_k B$ and $C_k$. The standardized results are as follows:
>
> | $k$                          | 1   | 2   | 3   | 4   | 5   |
> | ------------------------------ | --- | --- | --- | --- | --- |
> | **JNE($\lambda_k B$)** | .50 | .56 | .59 | .62 | .65 |
> | **JNE($C_k$)**         | .50 | .56 | .59 | .62 | .65 |
>
> The two curves are closely aligned, confirming that JNE reliably captures the degree of nonlinear modulation in the output space, consistent with our theoretical expectations.
>
> **Q2:** *On this same topic...*
>
> **Response:** We agree with the reviewer that the JNE score is influenced not only by intrinsic response properties of brain regions, but also by the diversity of input features and stimulus sets.
>
> To control for such confounding factors, we used the same CLIP-ViT representations and a unified natural image set across all brain regions, ensuring that JNE comparisons were conducted under consistent input conditions. This design allows us to focus specifically on regional differences in nonlinear response behavior, rather than input-driven variability in JNE scores.
>
> As the reviewer pointed out, changes in stimulus distribution (e.g., using only frontal faces) can affect JNE, even if the underlying brain function remains constant. This confirms that JNE reflects the input-dependent variability or output sensitivity of a region under a particular model and stimulus condition, rather than its absolute degree of nonlinearity.
>
> This conditional dependence is analogous to how $R^2$ also varies with stimulus diversity, yet remains a valid metric for comparing brain regions under the same experimental setup. In the revised manuscript, we will further clarify this conditional and relative nature of JNE, emphasizing that it should not be interpreted as an absolute or task-invariant measure of regional nonlinearity.
>
> **Q3:** *One more question on interpretation...*
>
> **Response:** As discussed in our response to W1, Q1 and Q2, the JNE value is jointly influenced by the structure of the model's mapping and the distribution of the input data. Moreover, it is further shaped by the nonlinear characteristics of the BOLD signal itself. Therefore, JNE should not be directly interpreted as a literal measure of the brain's intrinsic nonlinearity in response or processing. We will clarify this point explicitly in the revised manuscript.
>
> **Q4:** *I found myself wondering...*
>
> **Response:** We conducted a layer-wise analysis of JNE across all CLIP-ViT layers and found that JNE values vary systematically with model depth. We did not provide these results due to space limitations. In brief, JNE values increase sharply in the first two layers and stabilize in higher layers. Mid-to-late layers yield higher JNE scores in higher-order visual cortices, while early visual areas consistently show lower JNE values across all layers—typically below the whole-brain mean. Importantly, while the absolute JNE scores differ across layers, the relative ranking of brain regions remains highly consistent. This suggests that our core findings—such as hierarchical gradients and inter-regional differences—are robust to the choice of feature layer. The reviewer raised an insightful point that is worth further efforts. In the revision, we will focus on the methodological contribition of JNE in interpreting nonlinear encoding models.
>
> **Q5:** *In theory, with a powerful...*
>
> **Response:** In our response to **Q1**, we provided a theoretical argument suggesting that JNE can serve as an approximate proxy for nonlinearity in the target (output) space. We acknowledge that JNE is not a direct metric to measure the non-linearlity of the neural cortex, due to the complex nerual-vascular coupling mechanism.  Please refer our response to W1 and Q4 for details.
>
> **Q6.** *It would be helpful if the authors...*
>
> **Response:** We apologize for placing undue emphasis on the exploratory neuroscientific findings. In response, we will revise the manuscript to make the methodological contributions of this work more explicit. The fMRI encoding experiments serve as part of the validation instead. Additionally, we will incorporate prior studies on hierarchical processing in biological vision as the reviewer suggested. While we did observe some novel  patterns about the nonlinearity (e.g., category-specific nonlinearity preferences), the validation of those findings requires additional experiments involving multiple vision-language models, multiple fMRI datasets and more sophisticated statiscal comparisons.
>
> **Q7:** *If the goal is to characterize...*
>
> **Response:** JNE is not independent of the encoding model. As addressed in **Q1**, it serves as an approximate characterization of nonlinearity in the BOLD response space, conditioned on the given model. Furthermore, **Q4** demonstrates that the observed trends in JNE hold consistently across different representation layers, indicating that our conclusions are not sensitive to the choice of a specific layer.
>
> **Q8:** *In the Discussion...*
>
> **Response:** We thank the reviewer for their attention to this point. In the manuscript, our reference to a "bottom-up processing mechanism" is intended to describe the hierarchical progression by which visual information is processed from lower- to higher-level cortical areas, with increasing representational complexity. Specifically, we observed a monotonic increase in JNE values from early to higher visual regions, which supports a layered transition from linear to nonlinear processing along the cortical hierarchy. We will clarify this interpretation in the revised manuscript to avoid potential conceptual ambiguity.

---

> > ### Comment · Reviewer_VQtJ · 2025-08-05
> > **quality and rigor on methods clear, caution/clarity on interpretation needed**
> >
> > Thanks for your thoughtful responses. The quality and rigor in the methods seems clear to me, but I would re-iterate that caution/clarity on the interpretation is needed, and deeper integration with related neuroscience research would increase impact of the work. The authors seem to have a solid plan for both.

---

> ### Author Response · Authors · 2025-08-05
> **Response to Reviewer**
>
> We sincerely thank the reviewer for the thoughtful follow-up comment and for recognizing the quality and rigor of our methodology. We greatly appreciate your constructive guidance throughout the review process.
>
> As recommended, we will further emphasize caution and clarity in interpreting the JNE scores in the revised manuscript. Specifically, we will refine our claims to avoid overstating JNE as a direct measurement of neural-level nonlinearity. Instead, we will explicitly frame JNE as a model- and stimulus-conditioned diagnostic tool for characterizing the complexity of voxel-level mappings in nonlinear encoding models. We also acknowledge the importance of more deeply situating this work within the context of existing neuroscience literature. In the final revision, we plan to expand our discussion on hierarchical visual processing, neural-vascular coupling, and prior findings on brain-model correspondence. We will also clarify the methodological scope of JNE and identify future directions to strengthen its neuroscientific validity—such as evaluating consistency across model architectures and tasks.
>
> Once again, we appreciate your thoughtful review and suggestions, which have been invaluable in helping us improve the clarity and impact of this work.

---

### Official Review · Reviewer_gFGX · 2025-06-28

**Clarity:** 3
**Significance:** 2
**Originality:** 3
**Rating:** 5
**Confidence:** 4

**Summary:**

This paper introduces a measure (called JNE) to functionally compare output dimensions that are nonlinear functions of the input, focusing on nonlinear encoding models of biological sensory neurons. Using simple simulations, they show that JNE increases as model nonlinearities increase, and they also show JNE computed on models of visual cortical areas reflects hierarchical (and increasingly nonlinear) computations along visual areas.

**Questions:**

### Below are the questions for which I am happy to increase the rating given that they are convincingly addressed:

1\. How is the scale-dependence of the method justified? The current formulation conflates response gain with degree of nonlinearity, which is problematic: due to its scale dependence, the metric does not accurately reflect the intrinsic dimensionality or structure of curvature. This can yield identical scores for functionally distinct mappings through arbitrary rescaling—conflating response gain with actual nonlinearity. To illustrate this, consider two functions with inputs $x_1, x_2 \sim \mathcal{N}(0, 1)$:

* $f_1(x_1, x_2) = a x_1^2 \Rightarrow \nabla f_1 = [2a x_1,\ 0]$
* $f_2(x_1, x_2) = x_1^2 + x_2^2 \Rightarrow \nabla f_2 = [2x_1,\ 2x_2]$

Then the JNE (defined as the standard deviation of the L1 norm of the Jacobian across inputs) becomes:

* $\text{JNE}(f_1) = \mathrm{std}(|2a x_1|) = 2a \cdot \mathrm{std}(|x_1|)$
* $\text{JNE}(f_2) = \mathrm{std}(|2x_1| + |2x_2|) = 2 \cdot \mathrm{std}(|x_1| + |x_2|)$

Matching the JNEs yields: $ 2a \cdot \mathrm{std}(|x_1|) = 2 \cdot \mathrm{std}(|x_1| + |x_2|) \Rightarrow a = \frac{\mathrm{std}(|x_1| + |x_2|)}{\mathrm{std}(|x_1|)}$

This shows that **a 1D nonlinear function** can be made to appear **equally nonlinear** as a 2D one by simply amplifying its output. I am not convinced that response gain should be entangled with nonlinearity in a single metric, as they capture distinct computational aspects. This could be mitigated by standardizing Jacobian entries across inputs (e.g., subtracting mean and dividing by std), rather than just centering them. Why did the authors choose not to do this?

2\. Why use L1 norm and not L2? The proposed Jacobian-based metric aims to quantify a neuron's nonlinearity by aggregating input-specific gradients. However, for neurons with symmetric and rotationally invariant tuning profiles (i.e., invariance under rotations of the input space), the use of the L1 norm introduces a directional bias that breaks isoresponse symmetry. For example, consider a neuron with a 2D Gaussian tuning function: $f(x_1, x_2) = \exp\left(-\frac{x_1^2 + x_2^2}{2\sigma^2}\right)$. The gradient is $\nabla f = -\frac{f(x)}{\sigma^2} [x_1, x_2]$, and the **L2 norm** becomes: $\lVert \nabla f \lVert_2 = \frac{f(x)}{\sigma^2} \sqrt{x_1^2 + x_2^2}$. But since $x_1^2 + x_2^2 = -2\sigma^2 \log f(x)$, this yields: $\lVert \nabla f \lVert_2 = \sqrt{\frac{-2 f(x)^2 \log f(x)}{\sigma^2}}$, which depends only on the function value $f(x)$ — and thus is constant along isoresponse contours. In contrast, the **L1 norm** is: $\lVert\nabla f\lVert_1 = \frac{f(x)}{\sigma^2} (|x_1| + |x_2|)$, which varies across directions even when $f(x)$ is constant (i.e., on the same contour). In other words, equally nonlinear points are scored differently simply due to orientation. Using L2 could address this specific issue, and I wonder why the authors chose L1 norm.

3\. Why is the method "scale-normalized" as mentioned in the paper? which operation/step is related to this scale normalization? and why does that mean “the magnitude of standard deviation directly encodes biological sensitivity differences in neural responses”?

4\. I like that this approach is architecture-agnostic and can be applied on any model as long as the Jacobian matrix can be computed, but its dependence on the input introduces other challenges/questions: How and to what degree does JNE depend on the input distribution? How many inputs are needed? how to select the inputs? how does the number of inputs (and their selection) change with different number of output dimensions, and their degree of nonlinearity?

5\. While the method recovers the expected hierarchical trend in visual areas—consistent with prior findings—it is not immediately clear to me what new insights it enables beyond this. What does the metric reveal beyond "this area/neuron is more/less nonlinear than another"? The sample-specific analysis seems to hint at deeper insights, but ultimately still confirms known properties (e.g., early areas encode simple features). Could the authors clarify the kinds of new questions or discoveries this metric enables?

6\. Regarding the sample-specific analysis, I am not sure how much one can rely on or interpret these results. For instance, you state that “stimuli involving people or faces induce pronounced nonlinear fluctuations in FFA and EBA.” Looking at cluster 14 in Figure 12, this seems to hold—there are faces in the images, and FFA shows high MoM in the heatmap (Figure 8c). However, cluster 9 also contains many face stimuli, yet FFA does not exhibit similarly high MoM. Could the authors clarify this discrepancy or comment on the robustness of these observations?

---

### Additional questions
1\. The interpretation of JNE in terms of “stimulus sensitivity” was unclear to me. The authors write that "stimulus-sensitive regions (e.g., early visual cortex) exhibit greater parameter fluctuations, which mathematically aligns with the sample-dependent characteristics of nonlinear mechanisms" seemingly equating higher Jacobian variability (across input dimensions) and increased JNE. Could the authors clarify what does “parameter fluctuation” mean, in this context? and how does that “mathematically align with the sample-dependent characteristics of nonlinear mechanisms”?

2\. In figure 3d,e combination 2 and 3 have different JNE values, and you do mention it as an interesting observation but I could not find an explanation of why this happens. Intuitively in both cases you have linear → nonlinear → linear, and it is not clear to me why they should have different degrees of nonlinearity.

3\. In figure 7c, the caption says: where entry $(i, j)=1$ that means cortex i exhibits significantly lower JNE than cortex j. This means for instance HVC ($i=3$) exhibits significantly lower JNE than PVC ($j=1$). Shouldn’t it be the other way around? i.e. $(i, j)=1$ indicates statistically significantly “higher” as opposed to “lower”?

4\. What do the colors in figure 7d correspond to? Unfortunately, I did not get that from the caption. Is this a voxel-specific coloring or the voxels are based on the area label and the color is simply coming form the statistical test of difference between the areas? If it is the latter I am not sure what is the contribution of this figure exactly.

5\. I found the appendix defining JNE-SS a bit of an overkill. Isn’t JNE simply the square-root of JNE-SS averaged across stimuli? Changing the order to define JNE-SS first and then getting to JNE seems more intuitive to me, simply because JNE is an aggregate measure of JNE-SS, and maybe it could help simplify the definition/explanation.

**Ethical Concerns:**

["NO or VERY MINOR ethics concerns only"]

**Final Justification:**

The authors have provided a clear and well-argued response to my concerns, including a general derivation clarifying under what conditions JNE values can be equivalent across neurons, and why such equivalences are unlikely in practice. They also provided additional empirical validation and improved clarity around the metric’s design trade-offs. Overall, the authors' handling of these issues and the practical stability of the metric lead me to consider the contribution valuable and above the acceptance bar. I am therefore increasing my rating.

**Limitations:**

Some limitations are acknowledged by the authors and could serve as directions for future work. However, key limitations/questions—such as the method’s sensitivity to scale, and dependence on the input distribution were not discussed. These aspects are important for interpreting the utility and generalizability of the proposed metric and it is important to communicate them.

**Paper Formatting Concerns:**

None.

**Quality:**

2

**Strengths And Weaknesses:**

### Strength

- Technically sound derivation; tractable and well-motivated; reasonable evaluation on simulations and hierarchy
- Generally well-written, logical flow, readable
- Addresses an important topic—quantifying single-unit nonlinearity with a scalar, model-agnostic metric
- The proposed approach is novel in that it aims to provide a direct measure of nonlinearity that can be applied across different models, filling a gap not directly addressed by model comparison or geometry-based approaches.

### Weaknesses

- There are some conceptual issues (scale sensitivity, isoresponse inconsistency); some design choices are not clear (e.g. why L1 norm?)
- Some unclear phrasing ("stimulus sensitivity", “parameter fluctuations”); some definitions and the relation of different terms could be improved (e.g., JNE vs JNE-SS).
- Practical utility depends on resolving open issues (input distribution characteristics, scale sensitivity). The impact/significance of the method depends on how these are addressed.
- The interpretability of the sample-specific MoM analysis is unclear; some stimulus clusters with similar content (e.g., faces) elicit inconsistent responses in expected regions (e.g., FFA), raising questions about the robustness or specificity of the metric.
- Overlap with existing conceptual directions (intrinsic dimensionality, representational geometry); novelty could be better articulated relative to prior work.

---

> ### Author Rebuttal · Authors · 2025-07-31
>
> We would like to thank the reviewer for the constructive comments and suggestions, as well as the appreciation of our study. Our point-by-point responses are detailed as follows.
>
> ### **Response to Questions**
>
> **Q1:** *How is the scale-dependence of the method justified?*
>
> **Response:** We appreciate the reviewer's insightful example, which demonstrates that JNE is sensitive to output scale. However, it is important to clarify that JNE was not designed to quantify curvature in a purely mathematical sense, but rather to serve the interpretability of nonlinear neural encoding models. In this context, we intentionally preserve the magnitude information in the Jacobian standard deviation, based on the assumption that the *amplitude* of model-predicted responses may itself carry neurophysiological significance. For example, the overall responsiveness of a brain region or neuron to a stimulus may reflect biologically meaningful differences in sensitivity.
>
> Grounded in this motivation, we did not remove scale effects from the JNE formulation. Instead, we allow the absolute size of the Jacobian variability to reflect such sensitivity differences. Since the "structural space" of the input-output mapping is fixed during training, the response amplitude becomes an intrinsic part of the learned mapping. In other words, the model learns not only the shape of the curvature or representational manifold, but also its corresponding magnitude. Therefore, we argue that retaining response gain in the metric facilitates more direct links between the degree of nonlinearity and biological properties of the system, thereby enhancing physiological interpretability. This trade-off was a deliberate design choice in JNE: while we may sacrifice some theoretical purity in terms of scale-invariant curvature measurement, we gain a more direct characterization of real neural response behaviors. To further illustrate this, we note the following structural comparison between two functions:
>
> $$
> f_d(x) = \sum_{i=1}^d x_i^2, \quad f_{1D}(x) = a x_1^2,
> $$
>
> Under $x \sim \mathcal{N}(0, I_d)$, we have:
>
> $$
> \text{JNE}(f_d) = 2 \sqrt{d} \cdot \sigma_{|x|}, \quad \text{JNE}(f_{1D}) = 2a \cdot \sigma_{|x|} \Rightarrow a = \sqrt{d}.
> $$
>
> This result shows that to match the JNE value of a $d$-dimensional quadratic function, one must scale a single-dimension function by $a \sim \sqrt{d}$, e.g., $a \approx 28$ when $d = 768$. Such extreme amplification in a single input dimension is practically implausible in real neural encoding models, due to constraints from regularization, input-output normalization, parameter initialization, and activation saturation. These mechanisms jointly make it nearly impossible for a model to concentrate nonlinearity in a single direction at such scale.
>
> In the revised manuscript, we will explicitly state that the scale sensitivity of JNE is intentional, as it supports the interpretability of nonlinear responses in biological systems rather than undermining theoretical soundness.
>
> **Q2:** *Why use L1 norm?*
>
> **Response:** Our choice of the L1 norm was primarily motivated by its intuitive interpretability: it captures the overall response magnitude of a voxel to input perturbations via the sum of absolute gradient values across dimensions. This makes it more amenable to potential associations with neural sensitivity. In contrast, L2 norm, while geometrically symmetric, may obscure directional differences in local responses. It is important to emphasize that the goal of JNE is not to characterize the shape of iso-response profiles, but rather to compare the amplitude of nonlinear fluctuations across voxels (brain regions) under the same input perturbation distribution. Accordingly, we prioritize perturbation sensitivity, not geometric invariance. In addition, we tested the L2 norm and found that the overall spatial distribution pattern of JNE on the cortical surface was identical with that from L1, despite neumerical difference. We will clarify this design choice in the revised manuscript.
>
> **Q3:** *Why is the method "scale-normalized" ?...*
>
> **Response:** In both encoder training and JNE computation, we applied z-score normalization to the input features (i.e., CLIP representations) so that each input perturbation dimension had a standardized scale (0 mean, unit deviation). The target outputs of the encoder (fMRI beta values) were also z-scored prior to training, and both the model architecture and training protocol were kept consistent across voxels. Under this setting, the input space, output space, and mapping space (i.e., nonlinear neural encoding models) are all standardized. The normalized test inputs are likewise constrained within this normalized feature space, ensuring that both the training and evaluation phases operate under the same scale assumptions and statistical priors. As a result, the variability of the Jacobian can be directly interpreted as the relative sensitivity of each voxel to input perturbations, supporting comparisons across voxels and brain regions.
>
> **Q4:** *I like that this approach is architecture-agnostic ...*
>
> **Response:** We thank the reviewer for the appreciation of JNE. As a statistical measure, JNE is indeed influenced by the input distribution and sampling range. Sparse or skewed sampling may lead to an underestimation of the overall nonlinearity.
> To systematically assess its robustness, we conducted additional simulation experiments by varying both the sample size,  input scale and input distribution. Results show that even with as few as 30 samples, JNE can reliably works well under various input scales and distributions. Moreover, expanding the input scale from $[-100, 100]$ to $[-10^4, 10^4]$ produced highly stable JNE values, suggesting that the metric is primarily driven by model structure, rather than input scale alone.
> In the case of real fMRI data, we recognize that the stimulus set is often constrained by the experimental task, leading to non-ideal input distributions. Similar to $R^2$, we emphasize that JNE is a conditional metric—its value reflects nonlinear response characteristics under a specific model–data configuration, and are best interpreted in a relative sense across brain regions or models, rather than as an absolute measure of nonlinearity.
>
> In the revised manuscript, we will incorporate these results and discuss the limitation of JNE in interpreting real fMRI data.
>
> **Q5:** *While the method recovers the expected hierarchical ...*
>
> **Response:** The primary contribution of this work lies in the proposal and validation of the JNE metric for interpreting nonlinear encoding models.  While we did observe some interesting patterns (e.g., category-specific nonlinearity preferences), these were intended solely as preliminary demonstrations of the feasiablity of the proposed method. The validation of those findings requires additional experiments involving multiple vision-language models, multiple fMRI datasets and more sophisticated statiscal comparisons. In the revised manuscript, we will revise the wording to better emphasize the methodological contribution of our work.
>
> **Q6:** *Regarding the sample-specific analysis...*
>
> **Response:** In Cluster 9, the MoM value for FFA is higher than both the whole-brain average and early visual areas such as V1–V4 (see table below). However, since the figure applies max–min normalization within each region, this difference may be visually attenuated. In the revised manuscript, we will explicitly clarify this normalization procedure to avoid misinterpretation during cross-region comparisons.
>
> | Region | Brain | V1    | V2    | V3    | V4    | EBA   | PPA   | RSC   | OPA   | FFA-1 | FFA-2 |
> | ------ | ----- | ----- | ----- | ----- | ----- | ----- | ----- | ----- | ----- | ----- | ----- |
> | MoM    | 0.023 | 0.018 | 0.018 | 0.018 | 0.018 | 0.035 | 0.022 | 0.025 | 0.026 | 0.032 | 0.043 |
>
> ### **Response to Additional Questions**
>
> **AQ1:** *The interpretation of JNE in terms of...*
>
> **Response:** In the manuscript, the term "stimulus sensitivity" refers to the degree to which a brain region's Jacobian mapping depends on the input sample, i.e., the extent of variation across different visual stimuli. Similarly, "parameter fluctuation" denotes the variation of the Jacobian vector across input samples. These two terms essentially describe the same phenomenon: the variability of the model's Jacobian mapping in response to input perturbations. We will make these terms consistent in the revised manuscript.
>
> **AQ2:** *In figure 3d,e combination 2 and 3...*
>
> **Response:** Unfortunately, despite our best efforts to derive a formal explanation, the underlying mechanism remains unclear at this stage. We hypothesize that when nonlinearity is positioned in the later layers of a network, the longer gradient propagation path may increase the proportion of extreme Jacobian values, thereby elevating the JNE score. This pattern remains consistent across different activation functions and network architectures, suggesting that the position of nonlinearity may systematically affect its distribution. We plan to further investigate the underlying mechanism in future work and will design controlled experiments to validate this effect.
>
> **AQ3:** *In figure 7c, the caption says...*
>
> **Response:** Thank you for pointing this out and we will correct it accordingly.
>
> **AQ4:** *What do the colors in figure 7d correspond to?...*
>
> **Response:** You are absolutely correct: the colors in Figure 7d do not reflect voxel-level specificity, but are instead based on the JNE significance results aggregated at the region level. The purpose of this figure is to validate the effectiveness of the JNE metric, rather than to localize voxel-wise effects.
>
> **AQ5:** *I found the appendix defining JNE-SS...*
>
> **Response:** This is a great suggestion that help us to improve the clarity and readability of the manuscript. We will follow this suggestion to revise the manuscript.

---

> > ### Comment · Reviewer_gFGX · 2025-08-01
> > **Response to Author Rebuttal**
> >
> > Thanks for the detailed explanation and additional analysis.
> >
> > It helped me better understand that the goal of JNE is not to measure the degree of nonlinearity of a function per se, but to provide a functionally meaningful aggregate measure that enables comparisons between biological neurons, even when modeled with nonlinear mappings. From this perspective, the response scale (gain) is an important part of a neuron's functional signature, which is reflected in the proposed measure.
> >
> > While I agree that a metric for functional comparison across biological neurons should reflect scale, the issue that JNE conflates response gain, degree of nonlinearity, and number of input dimensions (that a given neuron is sensitive to) still remains, and can be problematic.
> >
> > Let us consider two cases:
> >
> > first, let’s consider the “plausibility” argument that you mentioned in your response, which is related to the entanglement between scaling factor and number of input dimensions. I agree with the authors that if the nonlinearity is concentrated in a single dimension, then one needs a large amplitude (gain) to match JNE, which is implausible. But if you consider another function with $k = d/2$ dimensions, then each dimension only needs to be scaled by $\sqrt{d/k} = \sqrt{2}$, which I’d say is plausible. In this case one function is sensitive to half of the dimensions compared to another, but just because its response is slightly higher (with a small scale) has the same JNR and is considered to be functionally the “same”.
> >
> > next, let us consider two neurons are responsive to the same number of input dims but one neuron is more nonlinear than another. Again in this case there exist a scale that makes these two neurons have the same JNE. Here is a very basic example:
> >
> > $f_1(x) = x^3, \ \ \ f_2(x) = ax^2$
> >
> > Under standard normal assumption:
> >
> > $\text{JNE}(f_1) = 3\cdot\sigma_{x^2} = 3\sqrt{2} \cdot \sigma_{x}^2 = \frac{3\sqrt{2}}{1 - 2/\pi} \cdot \sigma_{|x|}^2, \ \ \ \ \text{JNE}(f_2) = 2a\cdot\sigma_{|x|}
> > \Rightarrow{}
> > a = \frac{3\sqrt{2}}{2(1-2/\pi)} \cdot \sigma_{|x|}$
> >
> > Given the emphasize on the interpretability, I am wondering what do the authors think about the fact that two neurons with different levels of nonlinearity, or different level of dependency on the input dimensions, could yield similar JNE values?
> >
> > While I believe that these are aspects of the neuronal function that are independent and should ideally be measured and assessed independently, I agree that as far as a single measure is concerned, JNE can be useful. Overall, also considering that the authors response clarified other aspects, I am convinced that this work’s merits outweigh the theoretical limitations, and I am happy to increase my initial rating.

---

> ### Author Response · Authors · 2025-08-04
> **Response to Reviewer**
>
> We sincerely appreciate your positive feedback on our work, as well as your insightful mathematical analysis and theoretical questions. Your comments have significantly motivated us to reflect on and refine the fundamental interpretation of the JNE metric. We fully understand and agree with your core concern: JNE may be simultaneously influenced by response gain, the degree of nonlinearity, and dependencies across input dimensions, potentially confounding its interpretability.
>
> To address this issue, we have conducted a more systematic theoretical analysis in this revision and evaluated its impact through empirical results. Our analysis shows that under idealized conditions (e.g., when input derivatives are independent and identically distributed), JNE values for two neurons are equal if and only if their weights satisfy $w_1^\top \Sigma w_1 = w_2^\top \Sigma w_2$, where $\Sigma$ denotes the covariance matrix of the input gradient vector, and $w_1$, $w_2$ are the output projection vectors of the two neurons in the nonlinear encoding model.
>
> However, in real-world neural encoding models, the fully connected architecture typically induces complex dependencies across input dimensions. As a result, $\Sigma$ is generally a non-identity, non-diagonal matrix that includes substantial off-diagonal coupling terms. In such scenarios, even if $|w_1|_2 = \alpha |w_2|_2$, it does not guarantee a linear scaling relation $\text{JNE}_1 = \alpha \cdot \text{JNE}_2$, because the contribution of each input direction to JNE is modulated by the covariance structure.
>
> More critically, functional heterogeneity across neurons leads to divergent encoding goals, input sensitivities, and nonlinear response characteristics. These structural constraints imply that the output weights $w_1$ and $w_2$—learned through training—are unlikely to align in direction or magnitude, making the assumption $w_1 = \alpha w_2$ almost never satisfied in practice.
>
> In summary, JNE captures not only the scale of response gain but also the structural manner in which neurons organize nonlinear mappings across input dimensions. This directly aligns with our clarification in the revised manuscript: JNE is not merely a scalar measure of “nonlinearity strength,” but rather a functional metric that integrates both magnitude and structural attributes, aiming to support interpretable functional comparisons in neural system modeling. We will explicitly emphasize this point in the revised manuscript to directly address your critical concern.
>
> ---
>
> **Theoretical Derivation**
>
> In the nonlinear encoding model considered in our paper, for the $i$-th sample, the response functions of two neurons can be expressed as:
>
> $$
> f_{1,i}(x) =
> \begin{bmatrix}
> f(x_{1,i}) & f(x_{2,i}) & \cdots & f(x_{D,i})
> \end{bmatrix}
> \cdot w_1,\quad w_1 \in \mathbb{R}^{D \times 1}
> $$
>
> $$
> f_{2,i}(x) =
> \begin{bmatrix}
> f(x_{1,i}) & f(x_{2,i}) & \cdots & f(x_{D,i})
> \end{bmatrix}
> \cdot w_2,\quad w_2 \in \mathbb{R}^{D \times 1}
> $$
>
> where $D$ denotes the input dimensionality. The corresponding gradients are:
>
> $$
> \nabla f_{1,i}(x) =
> \begin{bmatrix}
> f'(x_{1,i}) & f'(x_{2,i}) & \cdots & f'(x_{D,i})
> \end{bmatrix}
> \cdot w_1
> $$
>
> $$
> \nabla f_{2,i}(x) =
> \begin{bmatrix}
> f'(x_{1,i}) & f'(x_{2,i}) & \cdots & f'(x_{D,i})
> \end{bmatrix}
> \cdot w_2
> $$
>
> Assuming $N$ samples and that each derivative $f'(x_{j,1\:N})$ has zero mean (as an idealized assumption to simplify analysis), we obtain:
>
> $$
> JNE_1 = \operatorname{std}\left(
> \left| \nabla f_{1,1}(x) \right|,
> \left| \nabla f_{1,2}(x) \right|,
> \cdots,
> \left| \nabla f_{1,N}(x) \right|
> \right)
> $$
>
> $$
> JNE_2 = \operatorname{std}\left(
> \left| \nabla f_{2,1}(x) \right|,
> \left| \nabla f_{2,2}(x) \right|,
> \cdots,
> \left| \nabla f_{2,N}(x) \right|
> \right)
> $$
>
> Let $\Sigma \in \mathbb{R}^{D \times D}$ denote the covariance matrix of the gradient vector $\begin{bmatrix} f'(x_1) & \cdots & f'(x_D) \end{bmatrix}^\top$, then:
>
> $$
> \text{JNE}_1 = \sqrt{1 - \frac{2}{\pi}} \cdot \sqrt{w_1^\top \Sigma w_1}
> \quad , \quad
> \text{JNE}_2 = \sqrt{1 - \frac{2}{\pi}} \cdot \sqrt{w_2^\top \Sigma w_2}
> $$
>
> Hence, $\text{JNE}_1 = \text{JNE}_2$holds if and only if **$w_1^\top \Sigma w_1 = w_2^\top \Sigma w_2$.**
>
> ---
>
> In recent days, we have also been actively exploring generalizations of JNE from a broader theoretical perspective, including efforts to analyze the interaction between nonlinear structure (e.g., curvature) and response gain in high-dimensional settings. However, due to the inherent complexity of nonlinear mappings in such spaces, this direction remains challenging. We plan to pursue this as a core focus in future work to further enhance the generality and theoretical robustness of the JNE framework.
>
> Once again, we sincerely agree with your core observations. Your theoretical insights have been deeply thought-provoking and inspiring. We greatly appreciate your precise feedback on both the mathematical formulation and the interpretability implications.

---

> ### Author Response · Authors · 2025-08-04
> **Response to Reviewer: Consistency Analysis of JNE for Real Data**
>
> To further assess the practical impact of this theoretical limitation—and incorporating your suggestion on normalization—we implemented and compared four JNE variants:
>
> * **JNE-L1**: The original version in our paper, using centered Jacobians with L1 norm;
> * **JNE-L2**: Replacing L1 with L2 norm;
> * **JNE-zscore-L1**: Applying z-score normalization across each input dimension, then computing the L1 norm;
> * **JNE-zscore-L2**: Same as above, but using the L2 norm.
>
> We applied these four JNE variants to real neural data to quantify regional nonlinearities, and computed pairwise Pearson correlation coefficients between the results of each variant:
>
> |                         | JNE-L1 | JNE-L2 | JNE-zscore-L1 | JNE-zscore-L2 |
> | ----------------------- | :----: | :----: | :-----------: | :-----------: |
> | **JNE-L1**        |   —   |  1.00  |     0.95     |     0.92     |
> | **JNE-L2**        |   —   |   —   |     0.94     |     0.92     |
> | **JNE-zscore-L1** |   —   |   —   |      —      |     0.99     |
> | **JNE-zscore-L2** |   —   |   —   |      —      |      —      |
>
> We observe that all versions exhibit high correlations (Pearson $r > 0.9$), indicating that the practical impact of theoretical sensitivity to response gain is limited. Final, we sincerely thank you once again for your valuable comments.

---

> > ### Comment · Reviewer_gFGX · 2025-08-07
> > **Final Remarks**
> >
> > Thank you for the detailed and thoughtful follow-up. I appreciate the additional derivation showing under what conditions JNE values could coincide across neurons, and your clarification that such cases are unlikely to occur in practice due to architectural and functional constraints.
> >
> > I also found the empirical comparison of JNE variants (L1 vs. L2, raw vs. z-scored) very useful, and the high correspondence between variants suggests that JNE is robust to these implementation choices.
> >
> > I would strongly recommend including both the derivation and the empirical comparison in the manuscript.
> >
> > With these clarifications and results, I feel more confident about the contribution and would recommend the paper for acceptance.

---

> > > ### Author Response · Authors · 2025-08-08
> > > **Final Response to Reviewer gFGX**
> > >
> > > We sincerely appreciate your positive feedback and recommendation for acceptance. Following your suggestions, we will include in the revised manuscript the theoretical derivation and empirical comparison of different JNE variants (L1/L2, raw/z-score) to further enhance the clarity and robustness of the method. Once again, thank you for your constructive comments and support throughout the review process.

---

### Official Review · Reviewer_m4zX · 2025-07-03

**Clarity:** 3
**Significance:** 3
**Originality:** 3
**Rating:** 5
**Confidence:** 3

**Summary:**

The key contribution of this paper is to provide a new metric for quantifying non-linearity in representations. In particular, they refer to this metric as Jacobian nonlinearity evaluation index (JNE). It measures how the Jacobian of the mapping from inputs to outputs changes over the input over—for a linear model this should be 0, while for a non-linear model we should see higher values for higher sensitivity to the space of inputs being considered. They apply this metric to study the amount of non-linearity in visual representations in different parts of the human visual cortex, by mapping embeddings of images to their voxel responses.

Aligning with previous results, they find that high-order visual orders have the most non-linearity, and that nonlinearity is hierarchically organized in the visual cortex. Separately, they also compute sample selectivity of representations in different regions and again consistent with literature they find that regions like FFA show pronounced non-linearity in response to faces (which they are known to be encoding).

**Questions:**

Following up on the weaknesses listed above:

1. I would like to know computational costs for these experiments to get a sense of how feasible they are to be used across domains.
2. While for the real experiments, it makes sense to stick to the data collected, it would be nice to know how sensitive the metric is to input samples in the simulations in order to strongly trust the real data results.
3. I am also curious about the choice of L1 norm here to measure deviation from the mean, and would like more clarification on that.

**Ethical Concerns:**

["NO or VERY MINOR ethics concerns only"]

**Final Justification:**

I am keeping my original score of 5. The paper was strong to start with, and the authors have thoughtfully answered my questions during the rebuttal phase.

**Limitations:**

yes

**Quality:**

3

**Strengths And Weaknesses:**

Strengths:

1. I absolutely love the figures and how the paper is written, everything is very clean and well-organized.
2. The metric is interesting, it is a novel way of quantifying non-linear between ANNs and brain encoding that goes beyond prediction performance.
3. Such mapping is widely studied in the human neuroscience community across modalities: audio, vision, etc and so this metric could be influential in extracting similar information about encodings in other parts of the brain.

Weaknesses:
1. This is minor but the limitation section should really go into the main text, and I'd urge the authors to do so if the paper gets accepted (in which case they will have an extra page).
2. Computing the Jacobian for a large model for various values of the input is a computationally expensive operation (as the authors detail in the limitations). The computation really depends on the input-output dimensionality, and size of sample space, which may restrict applicability.
3. A crucial part of computing this metric is sampling the right set and number of inputs over which the Jacobian is computed. This hasn't really been explored in the paper as of yet.

---

> ### Author Rebuttal · Authors · 2025-07-31
>
> We would like to thank the reviewer for the constructive comments and suggestions, as well as the appreciation of our study. Our point-by-point responses are detailed as follows.
>
> **W1:** *This is minor but the limitation section should really go into the main text...*
>
> **Response:** We will follow the reviewer's suggestion and move the "Limitations" section to the main text.
>
> **W2:** *Computing the Jacobian for a large model is computationally expensive...*
>
> **Response:**  Estimating the Jacobian of the models used in the current study via sample-wise backward computation, as traditionally done, may require 50 hours on a machine equipped with an Intel Xeon E5-2620 v4 CPU . To reduce this burden, our implementation employs an explicit forward-mode inference based on the structural formulation of the model. Specifically, we derive the closed-form Jacobian by analytically differentiating the linear transformations and activation functions at each network layer. This precedure relies solely on intermediate values and derivatives that are saved during the forward pass. As a result, this strategy reduces the computational time to 20–40 seconds, achieving an acceleration factor of over 99.9%.  We will provide implementation details and open-source the code. Below are the detailed formulation of the implimentation.
>
> For the nonlinear model used in our study, the forward structure is as follows:
>
> ```math
> h^{(0)} = \phi_1(W_1 x_i + b_1) \\
> h^{(1)} = \phi_2(W_2 h^{(0)} + b_2) + h^{(0)} \\
> h^{(2)} = \phi_3(W_3 h^{(1)} + b_3) + h^{(1)} \\
> \hat{y}_i = W_4 h^{(2)} + b_4
> ```
>
> Since commonly used activation functions (e.g., ReLU, GELU) have analytical first-order derivatives, we can record the sample-specific derivative matrices $W_{\phi_j, i}$ during forward propagation, and use them to calculate  the full Jacobian:
>
> ```math
> J_i = W_1 W_{\phi_1,i} W_2 W_{\phi_2,i} W_3 W_{\phi_3,i} W_4 \\
> + W_1 W_{\phi_1,i} W_3 W_{\phi_3,i} W_4 \\
> + W_1 W_{\phi_1,i} W_2 W_{\phi_2,i} W_4 \\
> + W_1 W_{\phi_1,i} W_4
> ```
>
> **W3:** *A crucial part of computing this metric is sampling the right set and number of inputs...*
>
> **Response:** The choice of input set and the number of samples indeed have a significant impact on the esitmation of JNE. To alleviate this problem, in the original submission, we applied multiple random sampling with averaging to reduce randomness and enhance statistical robustness in simulations. In real-data analysis, the training, validation, and test sets were randomly partitioned.
>
> To directly address the concerns such as "*Is the number of samples sufficient?*" and "*Is JNE sensitive to input perturbations?*", we inaddition designed a controlled simulation to quantitatively assess the impact of sample size. Our experimental results show that JNE is robust to input perturbations. When supplied with a reasonable number of samples (N>50), JNE can stably works well.
>
> The details of simulation experiments are as follows:
>
> 1. **Sample size variation**: We tested a range of sample sizes $N \in [10, 20, 30, 50, 100, 200, 300, 500, 1000]$;
> 2. **Signal perturbation factors**: We generated 1001 evenly spaced points in the interval $[0, 2\pi]$, defined a modulating signal $c = \sin(x)$, and used $|c|$ as an input perturbation factor;
> 3. **Jacobian simulation**: For each sample size $N$, we generated random tensors $J \in \mathbb{R}^{N \times 768 \times 1} \sim \mathcal{N}(0, 1)$, then modulated them pointwise using $c$ to obtain 1001 Jacobian sequences under varying input amplitudes;
> 4. **JNE computation**: We computed the JNE values corresponding to each modulated Jacobian sequence;
> 5. **Evaluation metric**: We measured the Pearson correlation coefficient between the resulting JNE curve and the target trend $|\sin(x)|$, to assess JNE's ability to track the input modulation.
>
> The results are summarized as follows:
>
> | Sample size$N$ | 10   | 20   | 30   | 50   | 100  | 200  | 300  | 500  | 1000 |
> | ---------------- | ---- | ---- | ---- | ---- | ---- | ---- | ---- | ---- | ---- |
> | Correlation$r$ | 0.87 | 0.94 | 0.96 | 0.97 | 0.99 | 1.00 | 1.00 | 1.00 | 1.00 |
>
> Even with as few as $N = 20$, the estimated JNE shows a strong and significant correlation with the target modulation pattern ($r = 0.87$). When $N \geq 100$, JNE almost perfectly fits the target curve.
>
> **Q1:** *I would like to know computational costs...*
>
> **Response:** Please see our response to W1.
>
> **Q2:** *It would be nice to know how sensitive the metric is to input samples...*
>
> **Response:** We aggree with the reviewer that evaluating the sensitivity of the JNE metric to input distributions is a critical step toward verifying its stability and generalizability. To this end, we extended the original simulation setup (Section 3.1) by systematically varying the input configuration along two dimensions—distribution type and numerical scale—to assess the robustness of JNE. Our experimental results demonstrate the high robustness of JNE.
>
> Simulation experiments are detailed as follows.
>
> 1. **Distribution type**: Inputs were sampled from either a uniform distribution $\mathcal{U}(a, b)$ or a normal distribution $\mathcal{N}(\mu, \sigma^2)$;
> 2. **Numerical scale**: Three amplitude ranges were tested—small $[-100, 100]$, medium $[-1000, 1000]$, and large $[-10000, 10000]$.
>
> Under each configuration, we evaluated JNE across four commonly used activation functions (ReLU, Leaky ReLU, GELU, and Swish), and four activation location settings (Loc Combination ∈ {1, 2, 3, 4}).
>
> **Mean JNE Values under *Uniform* Input Distributions**
>
> | Distribution Range   | $[-100, 100]$`<br>`Loc 1 | Loc 2 | Loc 3 | Loc 4 | $[-1000, 1000]$`<br>`Loc 1 | Loc 2 | Loc 3 | Loc 4 | $[-10000, 10000]$`<br>`Loc 1 | Loc 2 | Loc 3 | Loc 4 |
> | -------------------- | ---------------------------- | ----- | ----- | ----- | ------------------------------ | ----- | ----- | ----- | -------------------------------- | ----- | ----- | ----- |
> | **ReLU**       | 0.00                         | 0.12  | 0.14  | 0.16  | 0.00                           | 0.12  | 0.13  | 0.16  | 0.00                             | 0.12  | 0.14  | 0.16  |
> | **Leaky ReLU** | 0.00                         | 0.11  | 0.13  | 0.16  | 0.00                           | 0.12  | 0.14  | 0.16  | 0.00                             | 0.11  | 0.13  | 0.16  |
> | **GELU**       | 0.00                         | 0.12  | 0.14  | 0.16  | 0.00                           | 0.12  | 0.14  | 0.16  | 0.00                             | 0.12  | 0.14  | 0.16  |
> | **Swish**      | 0.00                         | 0.12  | 0.14  | 0.16  | 0.00                           | 0.12  | 0.14  | 0.16  | 0.00                             | 0.12  | 0.14  | 0.16  |
>
> These results are highly consistent across all tested ranges, indicating that JNE is insensitive to changes in input scale.
>
> **Mean JNE Values under *Normal* Input Distributions**
>
> | Distribution$\mathcal{N}(0, \sigma^2)$ | $\sigma^2 = 100$`<br>`Loc 1 | Loc 2 | Loc 3 | Loc 4 | $\sigma^2 = 1000$`<br>`Loc 1 | Loc 2 | Loc 3 | Loc 4 | $\sigma^2 = 10000$`<br>`Loc 1 | Loc 2 | Loc 3 | Loc 4 |
> | ---------------------------------------- | ------------------------------- | ----- | ----- | ----- | -------------------------------- | ----- | ----- | ----- | --------------------------------- | ----- | ----- | ----- |
> | **ReLU**                           | 0.00                            | 0.12  | 0.14  | 0.16  | 0.00                             | 0.12  | 0.14  | 0.16  | 0.00                              | 0.12  | 0.14  | 0.16  |
> | **Leaky ReLU**                     | 0.00                            | 0.12  | 0.13  | 0.16  | 0.00                             | 0.11  | 0.13  | 0.16  | 0.00                              | 0.11  | 0.13  | 0.16  |
> | **GELU**                           | 0.00                            | 0.12  | 0.14  | 0.16  | 0.00                             | 0.12  | 0.14  | 0.16  | 0.00                              | 0.12  | 0.14  | 0.16  |
> | **Swish**                          | 0.00                            | 0.12  | 0.14  | 0.16  | 0.00                             | 0.12  | 0.14  | 0.16  | 0.00                              | 0.12  | 0.14  | 0.16  |
>
> Under different variance configurations of $\mathcal{N}(0, \sigma^2)$, JNE remains almost unchanged, demonstrating the robustness of JNE to various distributions of inputs. These results suggest that JNE can reliably capture structural nonlinearity under a wide range of input settings.
>
> ***Q3:** The choice of L1 norm*
>
> **Response:** We adopted the L1 norm primarily due to its robustness and stability in measuring deviations from the mean in high-dimensional spaces. Compared to L2 norm, L1 norm computes absolute deviations, which mitigates the influence of extreme values. This makes it more effective in capturing the total magnitude of nonlinear perturbations, without being disproportionately affected by outliers.

---

> > ### Comment · Reviewer_m4zX · 2025-08-06
> > **post rebuttal**
> >
> > Thank you for your detailed response and additional experiments.
> >
> > The Jacobian computation makes sense, and I would encourage the authors to include this in the text. Also appreciate the experiments quantifying the number of samples and robustness to input distribution and again encourage the authors to add this to the supplement.
> >
> > I will retain my score and recommend an accept for this work.

---

> ### Author Response · Authors · 2025-08-08
> **Response to Reviewer m4zX**
>
> We sincerely thank you for your positive feedback on our additional experiments and implementation details. Following your suggestion, we will include the Jacobian computation method in the main text and add the detailed results of the sample size and input distribution robustness experiments to the appendix. We greatly appreciate your constructive comments and support throughout the review process.

---

### Decision · Program_Chairs · 2025-09-17

**Decision:**

Accept (spotlight)

**Comment:**

The reviewers and I agree that this is an interesting and novel work that is thorough and clearly presented. It suggests a useful perspective on how to quantify the nonlinearity in the (mapping from model representations to) neural representations, and demonstrates through simulations and analyses of real data that this metric behaves reasonably.

I find myself agreeing with the thoughtful interpretation of reviewer VQtJ about what nonlinearity is actually being quantified here (viz., the mapping itself), so I'd encourage the authors to take that perspective into account (as well as the remaining comments e.g. moving the limitations to the main text as promised), when preparing the camera ready.